# GPS: Graph-guided Proactive Information Seeking in Large Language Models

**Ruiqing Li**[1,2], **Yifeng Xu**[1,2], **Xinke Jiang**[1,2], **Zhibang Yang**[1,2], **Xinyu Ma**[1,2], **Yue Fang**[1,2],
**Junfeng Zhao**[2,3,6*], **Yasha Wang**[1,2,5*], **Xu Chu**[2,3,4*]

[1] National Engineering Research Center For Software Engineering, Peking University, Beijing, China
[2] School of Computer Science, Peking University, Beijing, China
[3] Key Laboratory of High Confidence Software Technologies, Ministry of Education, Beijing, China
[4] Center on Frontiers of Computing Studies, School of Computer Science, Peking University
[5] Peking University Information Technology Institute (Tianjin Binhai), Tianjin, China
[6] Big Data Technology Research Center, Nanhu Laboratory
lrq@stu.pku.edu.cn, chu_xu@pku.edu.cn

## ABSTRACT

Equipping Large Language Models (LLMs) with the ability to proactively ask clarifying questions is essential to mitigate ambiguity when faced with underspecified user queries in retrieval-augmented generation (RAG) systems. However, existing methods often neglect the rule-based reasoning structures embedded in the retrieved knowledge that are central to ambiguity, making it challenging to learn an effective and efficient question-asking strategy. To address these issues, we introduce **GPS**, a two-stage framework for enhancing proactive information seeking abilities of LLMs in RAG systems. In the reasoning stage, we propose a Directed Acyclic Graph (DAG) reasoning structure with theoretical guarantees of logical completeness, which facilitates capturing all conditional logic in the retrieved knowledge and supports effective clarification. In the clarification stage, we design a traversal-based algorithm that dynamically prunes the DAG based on user responses, enabling efficient clarification. To further enhance DAG construction, we first propose a conditional paths guided data synthesis method to address data scarcity challenge, then we apply a clarification-oriented reinforcement learning method with a hybrid reward that jointly considers effectiveness and efficiency to optimize the LLM. Experiments on three benchmarks demonstrate that **GPS** outperforms baseline methods in both success rate and clarification efficiency. [1]

## 1 INTRODUCTION

Consider a user seeking information about disability benefits eligibility and asks a question: "Am I eligible for disability premium?" While this question seems straightforward, the actual eligibility depends on multiple unstated conditions: income level, disability severity, and age. Without this critical information, even the most advanced retrieval-augmented generation (RAG) systems may provide incorrect or misleading answers. This scenario, illustrated in Figure 1, exemplifies a fundamental challenge in real-world question-answering systems: *how can AI systems proactively identify and gather missing information to provide accurate responses?*

This ambiguity primarily arises from underspecified user queries, which frequently occur in real-world

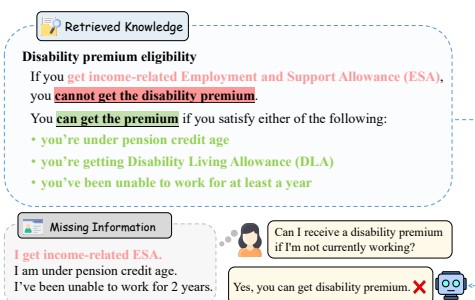

Figure 1: An illustration of an underspecified user query about disability benefits eligibility, where missing key information (e.g., income level, disability severity, age) may lead to incorrect LLM responses.

---

*Corresponding Author
[1]Our code is open-sourced at https://github.com/lrq111/GPS.git.

settings due to users' limited domain knowledge (Zhang et al., 2024; Deng et al., 2023a; Kim et al., 2024) or natural tendency to omit seemingly obvious details (Zipf, 1949). While existing RAG methods excel at retrieving relevant documents (Ren et al., 2025; Lewis et al., 2020; Zhao et al., 2025; Xu et al., 2025; Zhang et al., 2025b), they *fundamentally assume that user queries contain sufficient information, but this assumption often fails in practice*.

A promising solution is to equip Large Language Models (LLMs) in RAG systems with the ability to proactively ask clarifying questions when faced with underspecified queries. Currently, there are two main approaches: prompting and fine-tuning. Prompting methods (Deng et al., 2023b; Kuhn et al., 2023; Hu et al., 2024; Kobalczyk et al., 2025) utilize the reasoning capabilities of LLMs to iteratively identify ambiguity and generate clarification questions. However, their performance is constrained by the capability of LLMs as small-scale LLMs often struggle to identify ambiguities (Zhang et al., 2024). The better way is to fine-tune LLMs by multi-turn clarification dialogue data collected through human annotation (Qian et al., 2024; Chen et al., 2024) or self-sampling strategies (Andukuri et al., 2024; Zhang et al., 2025a). However, the former is costly to obtain, while the latter imposes no constraints on the clarification search space, potentially leading to irrelevant or redundant interactions. Therefore, it is necessary to develop an **effective** and **efficient** method to reach our goal.

We propose that the key to resolving ambiguity in underspecified queries lies in explicitly modeling the conditional reasoning structures within retrieved documents. Unlike existing methods that treat clarification as an open-ended dialogue problem, we observe that domain-specific documents typically encode knowledge as conditional rules—if-then statements that map combinations of conditions to conclusions. By extracting and representing these rules as a Directed Acyclic Graph (DAG), we can systematically identify all conditions relevant to the user's query and guide clarification dialogues through efficient traversal strategies.

However, realizing this vision presents three fundamental challenges. **(C1) How can we design a reasoning structure that captures all logical dependencies while remaining computationally tractable?** The structure must be expressive enough to represent arbitrary Boolean functions yet efficient enough for real-time interaction. **(C2) How can we train models to extract such structures when existing datasets lack annotations for conditional reasoning?** Current QA benchmarks rarely include underspecified queries or their missing conditions. **(C3) How can we optimize the extracted structures for both correctness and interaction efficiency?** Users will abandon systems that require excessive clarification rounds.

To address **(C1)**, we propose a conditional reasoning DAG structure, which is theoretically guaranteed to be logically complete to express any Boolean function via disjunctive normal form (DNF). Besides, the DAG allows for subgraph sharing across reasoning paths and supports dynamic pruning based on user responses, enabling $O(r)$ average-case clarification complexity, where $r \ll k$ is the average reasoning depth rather than the total number of conditions $k$. To address **(C2)**, we propose a conditional path guided data synthesis method to generate usable dataset for both training and evaluation. This method generates question-answer pairs with associated missing conditions along each conditional path from document. A filtering mechanism based on the necessity of the missing conditions is further applied to retain high-quality examples. To address **(C3)**, We propose a clarification-oriented reinforcement learning method to enhance LLM's ability to extract DAG structures for effective and efficient clarification. We design a hybrid reward that encourages the LLM to prioritize DAG that leads to correct answer and requires fewer interaction.

Our main contributions can be summarized as follows:

- **Novel Framework**: We introduce **GPS** (**G**raph-guided **P**roactive Information **S**eeking), the first framework to explicitly model conditional reasoning structures for clarification in RAG systems.
- **Theoretical Foundation**: We prove that our DAG-based representation achieves logical completeness while enabling $O(r)$ average-case clarification complexity, where $r \ll k$ is the average reasoning depth rather than the total number of conditions $k$.
- **Practical System**: We develop a complete pipeline including (i) Conditional path guided synthetic data generation to address training data scarcity, (ii) clarification-oriented reinforcement learning that jointly optimizes for accuracy and efficiency, and (iii) dynamic traversal algorithms that reduce user interaction burden.

- **Empirical Validation**: Extensive experiments on three benchmarks demonstrate that **GPS** achieves average improvement of **7.5% in success rate** and **4.2% in clarification efficiency** over the best baseline method.

## 2 RELATED WORK

**Clarification in LLMs**  Currently, there are two main approaches to enhance the ability of LLMs to proactively ask clarifying questions: prompting and fine-tuning. Prompting methods (Deng et al., 2023b; Kuhn et al., 2023) utilize the reasoning capabilities of LLMs to iteratively identify ambiguity based on the conversation history and choose to either ask clarification questions or generate response. However, their performance is constrained by the capability of LLMs as small-scale LLMs often struggle to identify ambiguities (Zhang et al., 2024), and as the conversation history grows longer, the risk of lost-in-the-middle increases (Liu et al., 2024). Another line of work is to fine-tune LLMs with multi-round conversation data (Qian et al., 2024; Chen et al., 2024). Yet these approaches rely on access to human-annotated conversation data, which is expensive to collect in practice. Some methods (Andukuri et al., 2024; Zhang et al., 2025a) explore self-improve paradigm for sampling conversation data and use the accuracy of final responses to filter low-quality clarification data. Nevertheless, these methods typically imposes no constraints on the clarification search space, potentially leading to irrelevant or redundant interactions. Therefore, it is necessary to develop an **effective** and **efficient** method for proactive clarification.

**Graph-based Reasoning in NLP**  Recent work has explored structured representations for multi-hop reasoning (Besta et al., 2024), knowledge graph integration (Ren et al., 2020; Li et al., 2025), and neural-symbolic reasoning (Xu et al., 2024). Query2Box (Ren et al., 2020) reasons over knowledge graphs by embedding multi-hop logical queries as geometric boxes in vector space. Li et al. (2025) proposes to inject LLMs with structured knowledge by encoding knowledge graphs via graph neural networks(GNNs). Xia et al. (2025) proposes a novel fine-tune framework stimulating the ability of LLMs to perform complex reasoning on knowledge graphs. However, these methods primarily encode entities, relations, or intermediate reasoning steps for well-specified queries, with the objective of improving multi-hop reasoning performance or the faithfulness of logical inference. Our work is explicitly designed for underspecified queries and uniquely combines graph-based reasoning with proactive clarification.

## 3 PROBLEM FORMULATION

In this paper, we aim to enhance LLMs' ability to proactively ask clarification questions when facing underspecified user query in RAG scenarios. Rigorously, given a user query $q$, the retrieved relevant document $d = Retrieve(q)$, and the user's background context $S$ which is not observable to the LLM, we denote by $C_d = \{c_1, \ldots, c_k\}$ the set of user-specific condition variables in $d$, each condition variable $c_i$ takes values from a finite value set $\mathcal{V}_{c_i}$.[2] We divide $C_d$ into two disjoint subsets:

- $C_{\text{known}}(q) \subseteq C_d$: the subset of **known condition variables** with values provided in query $q$.
- $C_{\text{miss}}(q) = C_d \setminus C_{\text{known}}(q)$: the subset of **missing condition variables** specific to query $q$, with values depend on the hidden user's background $S$ and are necessary to determine the answer.

Let $A = \{a_1, \ldots, a_m\}$ denote the set of possible answers. The final answer $a \in A$ is determined by $C_{\text{miss}}(q)$ through a set of latent logical constraints $\mathcal{R}$ encoded in $d$ (e.g., eligibility rules). Our objective is to enhance LLMs' ability to proactively elicit the values of $C_{\text{miss}}(q)$, so that an unambiguous answer $a$ can be inferred.

## 4 METHODOLOGY

In this section, we introduce **GPS**, a two-stage framework for proactive clarification. In the reasoning stage, a Reasoner LLM $\Theta_R$ captures the conditional structure in documents as a DAG. In the clarification stage, a Clarifier LLM $\Theta_C$ interacts with a User-Simulator LLM $\Theta_U$, dynamically

---

[2]For example, a condition variable $c_i$ = "marital status" may have $\mathcal{V}_{c_i} = \{\text{Single}, \text{Married}, \text{Divorced}\}$.

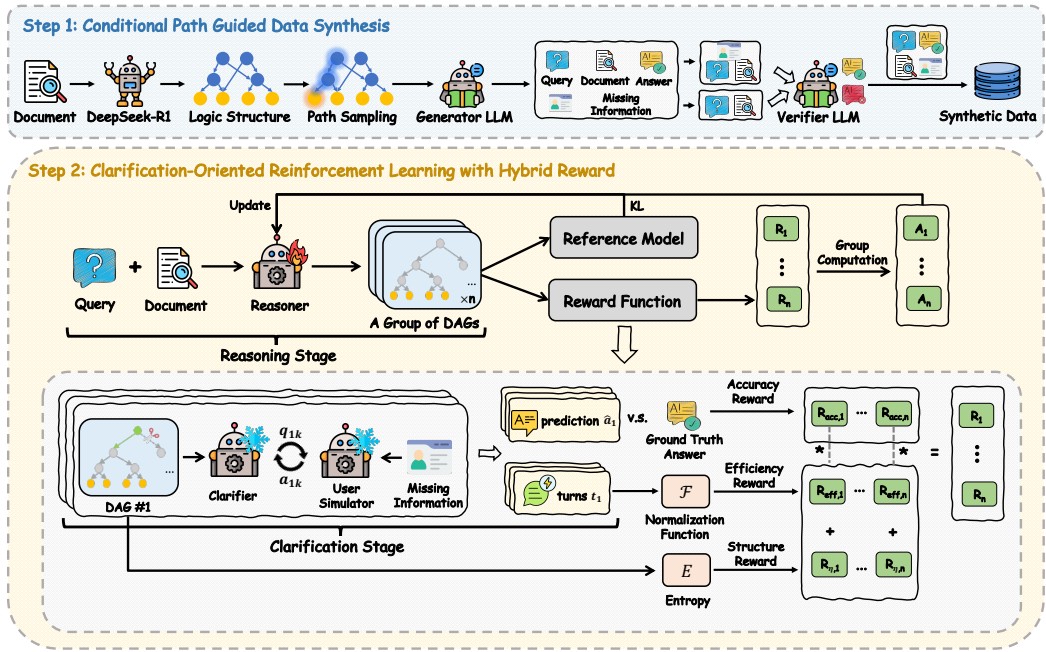

Figure 2: The overview of our method pipeline.

pruning the DAG during traversal to elicit values of $C_{\mathrm{miss}}(q)$. To further improve DAG quality, **GPS** employs the Conditional Path Guided Data Synthesis procedure to construct training dataset and utilizes Clarification-oriented Reinforcement Learning to optimize the Reasoner. The overall pipeline is illustrated in Figure 2.

## 4.1 CONDITIONAL REASONING DAG CONSTRUCTION

To construct a structure which is both logically complete and clarification-efficient, we define a conditional reasoning DAG structure $\mathcal{G} = (\mathcal{N}, \mathcal{E})$, where:

- Each non-terminal node $n_{c_i} \in \mathcal{N}$ represents a user condition variable $c_i \in C_d$, each terminal node $n_{a_m} \in \mathcal{N}$ represents a possible answer $a_m \in A$.

- Each edge $e_{i,j} = (n_{c_i}, n_{c_j}, \nu)$ is labeled with a condition value $\nu \in \mathcal{V}_{c_i}$, denoting a possible value of the predecessor node. The outgoing edges from each node are mutually exclusive and collectively exhaustive.

- If a node has a single predecessor node, it implicitly forms an **AND** relation with its predecessor condition. If a node has multiple predecessor nodes, it forms an **OR** relation over all predecessor conditions.

We first analyze the logical completeness of our proposed conditional reasoning DAG structure as follows. The clarification efficiency will be discussed in 4.2.

**Proposition 1.** *For any finite-valued function $g : \prod_{i=1}^{k} \mathcal{V}_i \to A$ over condition variables $\{c_i\}_{i=1}^{k}$, there exists a conditional reasoning DAG $\mathcal{G}$ such that, for each $a_m \in A$, every root-to-leaf path ending at $a_m$ corresponds to a conjunction in the disjunctive normal form (DNF) of the indicator function $\mathbf{1}[g(\cdot) = a_m]$, and the union of all such paths encodes the full DNF of $\mathbf{1}[g(\cdot) = a_m]$.*

See Appendix A for a complete proof. The proposition theoretically guarantees that our conditional reasoning DAG structure is expressive enough to represent any finite-valued function, allowing for comprehensive detection of missing conditions and effective clarification. In the reasoning stage, we prompt the Reasoner LLM $\Theta_R$ to construct a conditional reasoning DAG based on the user query and the retrieved document. The detailed prompt is provided in Appendix C.1.

## 4.2 Dynamic traversal-based clarification

Given the constructed conditional reasoning DAG $\mathcal{G} = (\mathcal{N}, \mathcal{E})$, we propose a dynamic traversal-based clarification approach that generates clarification questions in a topological order to prioritize essential conditions and dynamically prune inconsistent paths. Formally, let $\deg_{\text{in}}(n_i)$ denote the in-degree of node $n_i$, we define the candidate clarification set $U$:

$$
\begin{aligned}
U_1 &= \{n_i \in \mathcal{N} \mid \deg_{\text{in}}(n_i) = 0, \; i \notin C_{\text{known}}(q)\}, \\
U_2 &= \{n_i \in \mathcal{N} \mid \exists (n_u, n_i, \nu) \in \mathcal{E}, \; u \in C_{\text{known}}(q), \; \deg_{\text{in}}(n_u) = 0\}, \\
U &= U_1 \cup U_2
\end{aligned}
\tag{1}
$$

Nodes in $U$ are eligible for clarification as they are not blocked by unresolved predecessors. To determine the initial order of the candidate set, we estimate the expected cost based on the remaining depth for each $n_i \in U$:

$$
\ell(n_i) = \frac{1}{|P_{n_i}|} \sum_{p \in P_{n_i}} \text{len}(p)
\tag{2}
$$

where $P_{n_i}$ is a set of paths from $n_i$ to leaf node, and $\text{len}(p)$ is the number of missing condition variables in path $p$.

Our dynamic traversal-based clarification consists of three steps:

1. **Initialization:** construct the candidate set of missing conditions and start with empty dialogue history $H$;
2. **Iterative clarification:** select the most informative condition, ask a clarification question via $\Theta_C$, obtain a user-specific answer from $\Theta_U$, and update $H$ by following the consistent path;
3. **Final answering:** once a terminal condition is reached or no candidates remain, $\Theta_C$ produces the final answer $\hat{a}$ conditioned on $H$.

Through dynamic traversal-based clarification process, the dialogue history $H$ constitutes the elicited values for the missing condition set $C_{\text{miss}}$, which are required to resolve the ambiguity. We present the algorithmic pseudo-code of dynamic traversal-based clarification in Appendix D. We also analyze the efficiency of GPS and obtain that the expected number of clarification turns depends only on the small set of conditions along the true reasoning path, rather than the full set of conditions present in the document. Detailed analysis is provided in the Appendix B.

## 4.3 Clarification-oriented Reinforcement Learning

Enhancing the Reasoner LLM's ability to construct accurate conditional reasoning DAG is essential for the overall performance of the **GPS** framework. To achieve this, we first propose a *Conditional Path Guided Data Synthesis* procedure to address the data scarcity challenge. Based on the synthetic dataset, we design a *Clarification-Oriented Reinforcement Learning* approach with a hybrid reward that integrates clarification effectiveness and efficiency, encouraging the Reasoner LLM to extract DAG structures that lead to correct answer and require fewer interaction.

### 4.3.1 Conditional path guided data synthesis

We synthesize our dataset based on ConditionalQA (Sun et al., 2022), a reading comprehension dataset that includes long-context documents containing complex logic rules, along with well-specified and underspecified queries with human-annotated missing conditions. However, a key limitation of ConditionalQA dataset is that only 550 out of 2,247 samples (24.5%) are underspecified, making it difficult to train models that generalize well in proactive information seeking task. To address this, we propose *Conditional Path Guided Data Synthesis* method to augment high-quality underspecified training data. The synthesis process consists of two steps: **Problem Generation** and **Verification**.

**Problem Generation** First, we prompt advanced LLMs such as DeepSeek-R1 (Guo et al., 2025) to generate underspecified questions with multi-conditional reasoning paths from a document $d$. The prompt used for this task can be found in Appendix C.2. Each item contains three parts:

- An **underspecified question** $q$ that admits multiple plausible answers;

- A set of **missing conditions** $C$ with value domains $\{\mathcal{V}_c\}_{c \in C}$;
- A set of **conditional paths** $\mathcal{P} = \{(\mathbf{v}, a)\}$, where $\mathbf{v} \in \prod_{c \in C} \mathcal{V}_c$ is a complete assignment, $a \in A$ is the *unique* answer determined by $\mathbf{v}$.

**Verification** Each synthetic data instance is represented as $(q_i, \mathbf{v}_i, d_i, a_i)$. To ensure data quality, we introduce a filtering mechanism based on the necessity of the missing conditions. For each instance, we prompt a Verifier LLM to predict the answer both with and without access to the missing conditions. We retain the instance only if the full-input prediction $a_{\text{full}}$ matches the gold answer $a_i$, while the masked-input prediction $a_{\text{partial}}$ does not. This filtering preserves cases where missing information is essential, yielding a high-quality dataset $\mathcal{D} = \{q_i, \mathbf{v}_i, d_i, a_i\}_{i=1}^n$.

### 4.3.2 HYBRID REWARD: EFFECTIVENESS, EFFICIENCY, AND ENTROPY

**Setup.** Our goal is to train the *Reasoner* LLM $\Theta_R$ to generate high-quality conditional reasoning DAGs that enable a *fixed* Clarifier LLM $\Theta_C$ to complete proactive clarification with minimal interaction. Motivated by recent progress of RL for reasoning(Guo et al., 2025), we formulate the DAG-construction problem as a reinforcement learning task: the **policy** $\pi_\theta(o \mid q, d)$ autoregressively generates and parses a DAG $o$ given query $q$ and document $d$ (we write $\theta$ as the Reasoner parameters $\Theta_R$ for brevity), the DAG deterministically induces a clarification trajectory (which condition to ask next and which edge to follow), and yielding a scalar **reward**. We adopt GRPO (Guo et al., 2025) algorithm to optimize $\pi_\theta$.

**GRPO objective.** For each query $q$ and retrieved document $d$, GRPO samples $h$ DAGs $\{o_i\}_{i=1}^h \sim \pi_{\theta_{\text{old}}}(\cdot \mid q, d)$ with rewards $\{r_i\}_{i=1}^h$, defines the advantage $A_i$ as

$$A_i = \frac{r_i - \text{mean}(\{r_1, \ldots, r_h\})}{\text{std}(\{r_1, \ldots, r_h\})}, \tag{3}$$

and optimizes the clipped policy objective with a reference KL:

$$\mathcal{J}_{\text{GRPO}}(\Theta) = \mathbb{E}_{q \sim D, \, \{o_i\} \sim \pi_{\theta_{\text{old}}}(\cdot \mid q)} \frac{1}{h} \sum_{i=1}^h \Big[ \min\Big( \frac{\pi_\theta(o_i \mid q)}{\pi_{\theta_{\text{old}}}(o_i \mid q)} A_i, \, \text{clip}\big(\frac{\pi_\theta}{\pi_{\theta_{\text{old}}}}, \, 1 - \epsilon, \, 1 + \epsilon\big) A_i \Big) \tag{4}$$
$$- \, \beta \, \mathbb{D}_{\text{KL}}\big(\pi_\theta \, \| \, \pi_{\text{ref}}\big) \Big],$$

with

$$\mathbb{D}_{\text{KL}}\big(\pi_\theta \, \| \, \pi_{\text{ref}}\big) = \frac{\pi_{\text{ref}}(o_i \mid q)}{\pi_\theta(o_i \mid q)} - \log \frac{\pi_{\text{ref}}(o_i \mid q)}{\pi_\theta(o_i \mid q)} - 1. \tag{5}$$

GRPO calculates *group-relative* advantages within $h$ samples $\{o_i\}_{i=1}^h$ for the same input, so higher-reward generated DAG samples are upweighted and lower-reward ones are downweighted.

**Hybrid reward function.** To optimize the policy toward high-quality DAGs, we design a hybrid reward function that jointly promotes correctness, low interaction cost, and structurally non-redundant.

- **Effectiveness reward** encourages correct final answers after clarification. Let $\hat{a}_i$ be the predicted answer after clarification based on DAG $o_i$ and $a_i$ the ground truth:

$$r_{\text{acc},i} = \begin{cases} 1, & \texttt{Evaluator}(\hat{a}_i, a_i) = \texttt{True}, \\ 0, & \texttt{Evaluator}(\hat{a}_i, a_i) = \texttt{False}. \end{cases} \tag{6}$$

- **Efficiency reward** encourages fewer clarification turns. Let $t_i$ be the number of turns induced by $o_i$, and $t_{\max} = \max_{j \in [1,h]} t_j$ within the GRPO group. We set the coefficient $\alpha = 0.5$:

$$r_{\text{eff},i} = 1 - \alpha \frac{t_i}{t_{\max}}, \tag{7}$$

- **Structural quality reward** $r_{\eta,i}$ measures how effectively a DAG converts intermediate branching uncertainty into discriminative power over the final conclusions. We formalize this intuition using an information–conversion efficiency measure.

**Forward probability propagation.** Let $\mathcal{G} = (\mathcal{N}, \mathcal{E})$ be a clarification DAG with condition nodes $C$ and conclusion (leaf) nodes $L$. We assume *uniform branching*: at any condition node, the outgoing probability mass is evenly split among all children. Let $R \subseteq C$ be condition roots with no predecessors. Each root receives initial mass $1/|R|$. For any condition node $n \in C$, its outgoing mass splits uniformly across children:

$$P(v) = \sum_{u:(u \to v) \in \mathcal{E}} \frac{P(u)}{|F(u)|}, \tag{8}$$

where $F(u)$ is the set of children of $u$.

This yields a forward-reachability distribution $P(n)$ over all nodes. We can obtain the mass $P(\ell)$ of any leaf node $\ell \in L$ and the normalized leaf distribution is:

$$\tilde{P}(\ell) = \frac{P(\ell)}{\sum_{\ell' \in L} P(\ell')}. \tag{9}$$

**Entropy of graph splits and leaf.** The *graph split entropy* reflects the total uncertainty injected by intermediate splits:

$$H_{\text{graph}} = \sum_{n \in C} P(n) \log |F(n)|, \tag{10}$$

and the *leaf entropy* is the Shannon entropy of the normalized leaf distribution:

$$H_{\text{leaf}} = -\sum_{\ell \in L} \tilde{P}(\ell) \log \tilde{P}(\ell). \tag{11}$$

**Information-conversion efficiency.** We define the structural quality reward $r_{\eta,i}$ as

$$r_{\eta,i} = \begin{cases} \dfrac{H_{\text{leaf}}}{H_{\text{graph}}}, & H_{\text{graph}} > 0, \\ 1, & H_{\text{graph}} = 0 \ \wedge \ \exists \ell \in L : P(\ell) > 0, \\ 0, & \text{otherwise.} \end{cases} \tag{12}$$

We provide illustrative instances in Appendix G.3, demonstrating the rationality of the structural quality reward.

- **Overall reward.** The $i_{th}$ sample's total reward is calculated as follows:

$$r_i = r_{acc_i} \cdot (r_{eff_i} + r_{\eta,i}) \tag{13}$$

We provide the pseudo-code of overall inference procedure of GPS in Appendix E and training procedure of the Reasoner in Appendix F, and additionally offer an illustrative example of the two-stage proactive information-seeking process in Appendix H.

## 5 EXPERIMENT

### 5.1 EXPERIMENTAL SETUP

**Dataset** We constructed the GPS training dataset based on ConditionalQA (Sun et al., 2022) and we test our method on the following three datasets. The detailed statistics of the datasets are introduced in Appendix K.

- **Synthetic** is the test split of our conditional path guided synthetic dataset, consisting entirely of underspecified queries.
- **ConditionalQA** (Sun et al., 2022) includes both well-specified queries and underspecified queries. It provides annotation for each question-answer pair along with the document and the corresponding missing conditions. For well-specified queries, the missing conditions are empty.
- **ShARC** (Verma et al., 2020) is a conversational QA dataset that also includes well-specified queries and underspecified queries based on rules expressed in natural language text. For underspecified queries, it provides annotated clarification dialogues. We concatenate the clarification dialogue as the missing conditions. Compared to typical datasets in RAG scenarios, ShARC features much shorter documents and a restricted answer space limited to *yes* or *no*. We use ShARC to evaluate the generalization ability of our method.

**Baselines** We adopt the following state-of-the-art approaches as our compared baselines.

- **Base Method** answers user query directly based on the relevant document, which can be considered as fundamental framework in RAG.

- **ProCoT** (Deng et al., 2023b) leverages Chain of Thought prompting scheme to judge whether the user query is underspecified and generate a clarification question if needed.

- **UoT** (Hu et al., 2024) proposes Uncertainty of Thought prompting, which enhances LLM reasoning by explicitly modeling and reducing uncertainty during the reasoning process.

- **BED-LLM** (Kobalczyk et al., 2025) uses Bayesian Experimental Design to pick the clarification question that maximizes information gain.

- **Adaptive-BED-LLM** is an ambiguity-adaptive variant of BED-LLM for fairer comparison, as the original method always chooses to ask clarification questions. It answers directly when multiple initial answers are semantically consistent, otherwise performs BED-based clarification.

- **Clarify-DPO** (Zhang et al., 2025a) is a fine-tuning based method. It leverages a self-improve method to collect training data and filter data by gold answer.

- **Adaptive-Clarify-DPO** is an ambiguity-adaptive variant of Clarify-DPO based on the *Clarify-or-Direct Answer* strategy proposed in (Zhang et al., 2025a). The model learns to choose either generating a clarification question or directly answering.

**Models** We evaluate the performance using Llama3-8B-Instruct (Grattafiori et al., 2024) and Qwen2.5-7B-Instruct (Yang et al., 2025) as backbone models.

**Evaluation Metrics** We evaluate the model's proactive information seeking ability using the following four metrics:

- **Success Rate (SR)**. Following previous studies(Hu et al., 2024; Qian et al., 2024), we use this metric measures the **effectiveness** of clarification process by computing the proportion of the correct predictions after clarification. We employ an evaluator LLM to judge the semantic alignment between the predicted answer and the ground-truth answer. The evaluation prompt is provided in Appendix C.3.

- **Weighted Clarification Turns (WCT).** Previous studies commonly evaluate clarification efficiency using Mean Clarification Turns (MCT), defined as the average number of clarification questions (Hu et al., 2024; Qian et al., 2024). However, the desired behavior of a proactive information seeking model is to prioritize correct clarification before optimizing efficiency, and MCT alone cannot capture **success-conditioned efficiency**. Inspired by prior evaluation protocols (Yokoyama et al., 2021), we introduce the Weighted Clarification Turns (WCT):

$$\text{WCT} = p_{\text{success}} \cdot \text{MCT}_{\text{success}} + p_{\text{failed}} \cdot T_{\text{max}}, \tag{14}$$

where $p_{\text{success}}$ and $p_{\text{failed}}$ denote the proportions of successful and failed samples, $\text{MCT}_{\text{success}}$ is the mean clarification turns over successful samples, and $T_{\text{max}} = 10$ is the maximum clarification-turn budget in our experiments. Lower WCT indicates more efficient clarification while preserving correctness.

- **F1 score** for Clarification Need Prediction Accuracy (CNP). Following previous studies (Deng et al., 2023b; Zhang et al., 2025a), we compute the F1 score of CNP for evaluating the model's ability to identify the necessity of clarification.

## 5.2 PERFORMANCE COMPARISON

Table 1 presents the performance comparison of different methods across three benchmarks: Synthetic, ConditionalQA, and ShARC. We summarize key findings below:

**Training for proactive information seeking is essential.** The Base Method yields low success rates on Synthetic and ShARC, where underspecified queries are more prevalent, indicating its limited ability to handle missing conditions. Baselines equipped with proactive clarification consistently improve SR over the Base Method. However, purely prompt-based methods sometimes fail to surpass the Base Method. For example, ProCoT occasionally results in degraded performance, likely

Table 1: **Performance comparison on three datasets.** Columns report SR (Success Rate, %), Weighted Clarification Turns (WCT), and F1 score (%). **Bold** indicates the best result, while underline denotes the second-best results.

| Method | Synthetic | | | ConditionalQA | | | ShARC | | |
|---|---|---|---|---|---|---|---|---|---|
| | SR (↑) | WCT (↓) | F1 (↑) | SR (↑) | WCT (↓) | F1 (↑) | SR (↑) | WCT (↓) | F1 (↑) |
| *Qwen2.5-7B-Instruct* | | | | | | | | | |
| Base Method | 21.2 | 7.88 | 0.0 | 70.3 | 2.98 | 0.0 | 49.3 | 5.08 | 0.0 |
| ProCoT | 42.5 | 6.07 | 50.9 | 71.6 | 2.95 | 10.4 | 62.6 | 4.06 | 51.3 |
| UoT | 32.8 | 7.05 | 89.2 | 60.3 | 4.25 | 28.2 | 70.5 | 3.25 | 83.8 |
| BED-LLM | 40.9 | 6.41 | **100.0** | 52.8 | 5.26 | **37.6** | 62.2 | 4.22 | 66.7 |
| Adaptive-BED-LLM | 34.6 | 6.89 | 95.2 | 50.2 | 5.58 | 28.6 | 59.4 | 4.56 | 71.0 |
| Clarify-DPO | 59.2 | 4.67 | **100.0** | 72.0 | 3.52 | **37.6** | 78.5 | 2.93 | 66.7 |
| Adaptive-Clarify-DPO | 32.6 | 7.04 | 96.2 | 69.9 | 3.02 | 0.0 | 70.0 | 2.99 | 0.0 |
| **GPS** | **60.2** | **4.59** | 96.4 | **73.4** | **2.91** | 36.7 | **79.3** | **2.41** | **87.5** |
| *LLaMA3-8B-Instruct* | | | | | | | | | |
| Base Method | 30.8 | 6.92 | 0.0 | 62.8 | 3.72 | 0.0 | 56.6 | 4.34 | 0.0 |
| ProCoT | 28.3 | 7.62 | 29.2 | 66.3 | 3.58 | 25.7 | 53.7 | 5.16 | 35.6 |
| UoT | 29.7 | 7.36 | 90.9 | 64.6 | 4.03 | 31.7 | 68.3 | 3.67 | 69.9 |
| BED-LLM | 39.6 | 6.53 | **100.0** | 47.2 | 6.02 | **37.6** | 64.0 | 4.20 | 66.7 |
| Adaptive-BED-LLM | 35.6 | 6.84 | 96.2 | 44.5 | 5.90 | 31.4 | 67.8 | 3.49 | 67.0 |
| Clarify-DPO | 53.2 | 5.21 | **100.0** | 66.3 | 4.03 | **37.6** | **82.7** | **2.55** | 66.7 |
| Adaptive-Clarify-DPO | 31.3 | 7.16 | 95.5 | 67.7 | 3.23 | 0.0 | 68.8 | 3.12 | 0.0 |
| **GPS** | **56.5** | **5.02** | 96.2 | **74.6** | **2.89** | 28.0 | 75.8 | 2.79 | **82.5** |

due to limited backbone capacity and the inherent complexity of conditional reasoning, consistent with prior observations (Zhang et al., 2024).

**GPS achieves the best balance between effectiveness and efficiency.** GPS consistently improves SR across baselines. With LLaMA-3-8B-Instruct, GPS achieves an average relative SR gain of 10.4% over the second-best method, and 4.5% with Qwen2.5-7B-Instruct. Moreover, under WCT, which jointly reflects correctness and efficiency, GPS attains the lowest WCT in nearly all settings, indicating higher accuracy at lower effective clarification cost.

**Strong generalization to ShARC.** GPS also generalizes well to the ShARC dataset, where it consistently outperforms prompt-based methods and Base Method, and achieves performance comparable to the Clarify-DPO method, despite the fact that Clarify-DPO is trained directly on ShARC. This highlights the strong generalization ability of GPS across different benchmarks.

## 5.3 ABLATION STUDY

To evaluate the contribution of each component in GPS, we conduct an ablation study using Qwen2.5-7B-Instruct as the backbone LLM. Results are reported in Table 2 on both the Synthetic and ConditionalQA datasets.

The full GPS model achieves the best overall performance across both datasets. Removing reinforcement learning (w/o RL) leads to a notable drop in SR and a clear increase in WCT, particularly on the Synthetic dataset, highlighting the importance of policy optimization. Ablating either the Efficient Reward or the Structural Quality Reward consistently degrades performance, increasing WCT and reducing SR, which indicates that jointly modeling correctness and efficiency is crucial. Finally, disabling dynamic traversal results in higher WCT on both datasets, suggesting its role in reducing effective clarification cost by guiding more efficient clarification paths.

## 5.4 QUALITATIVE ANALYSIS

Figure 3 presents comparison between a strong baseline Clarify-DPO and our method GPS on an underspecified query about policy eligibility. Clarify-DPO selects clarification questions based on

Table 2: Ablation results of GPS with Qwen2.5-7B-Instruct.

| Method | Synthetic | | | ConditionalQA | | |
|---|---|---|---|---|---|---|
| | SR↑ | WCT↓ | F1↑ | SR↑ | WCT↓ | F1↑ |
| GPS | 60.2 | 4.59 | 96.4 | 73.4 | 2.91 | 36.7 |
| w/o RL | 52.2 | 5.57 | 96.8 | 67.7 | 3.63 | 23.1 |
| w/o Efficient Reward | 59.0 | 5.06 | 97.1 | 70.7 | 3.58 | 28.9 |
| w/o Structural Quality Reward | 56.1 | 5.32 | 96.3 | 70.3 | 3.61 | 29.1 |
| w/o Dynamic Traversal | 59.6 | 5.19 | 96.4 | 71.2 | 3.63 | 22.2 |

implicit reasoning, which leads to missing essential condition about income-related ESA, and the resulting clarification path does not cover all necessary branches. This omission causes the model to produce an incorrect final answer.

In contrast, GPS first constructs a conditional reasoning DAG that explicitly enumerates all relevant conditions. The dynamic traversal module then identifies the most informative condition to clarify, removes inconsistent branches based on user responses, and narrows the search to the uniquely valid leaf node. This produces a clarification trajectory that is both minimal and logically complete. For more qualitative analysis, please refer to Appendix G.

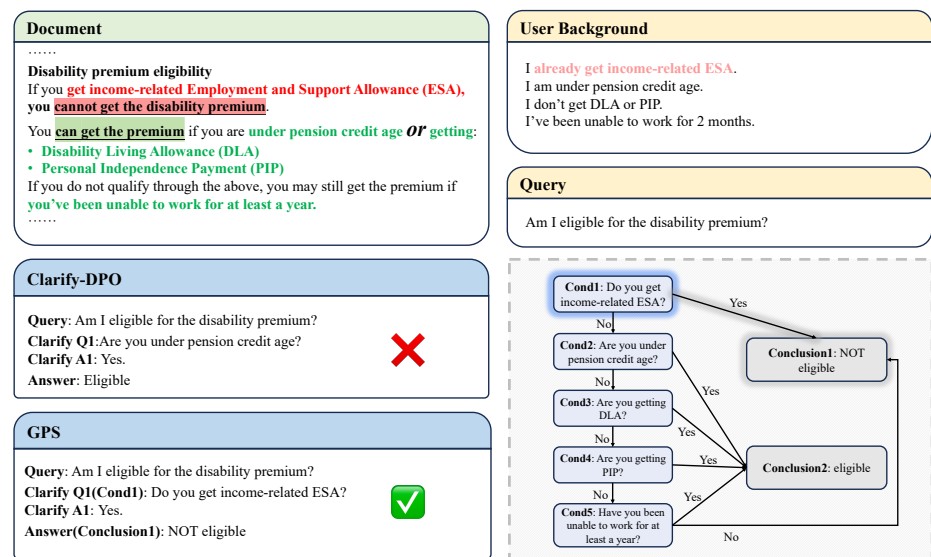

Figure 3: Comparison of Clarify-DPO and GPS on a policy eligibility example. Clarify-DPO asks an incomplete set of clarification questions and reaches an incorrect answer. GPS constructs a conditional reasoning DAG and identifies the correct clarification path, producing the correct conclusion.

## 6 CONCLUSION

In this paper, we propose GPS, a two-stage framework for enhancing proactive information seeking abilities of LLMs in RAG systems. In the reasoning stage, we propose a DAG reasoning structure with theoretical guarantees of both logical completeness and clarification efficiency. In the clarification stage, we design a traversal-based algorithm that dynamically prunes the DAG based on user responses, enabling efficient clarification. To further enhance DAG construction, we propose a conditional path guided data synthesis method to address data scarcity challenge, then we apply a clarification-oriented reinforcement learning method with a hybrid reward that jointly considers effectiveness and efficiency to optimize the LLM. Extensive experiments on three benchmarks demonstrate the effectiveness and efficiency of GPS in handling underspecified queries.

ACKNOWLEDGMENTS

This work is supported by the National Natural Science Foundation of China (No. 62576013).

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

## A  PROOF OF PROPOSITION 1

**Proposition 1.** *For any finite-valued function $g : \prod_{i=1}^{k} \mathcal{V}_i \to A$ over condition variables $\{c_i\}_{i=1}^{k}$, there exists a conditional reasoning DAG $\mathcal{G}$ such that, for each $a_m \in A$, every root-to-leaf path ending at $a_m$ corresponds to a conjunction in the disjunctive normal form (DNF) of the indicator function $\mathbf{1}[g(\cdot) = a_m]$, and the union of all such paths encodes the full DNF of $\mathbf{1}[g(\cdot) = a_m]$.*

*Proof.* Let $g : \prod_{i=1}^{k} \mathcal{V}_i \to A$ be a total function over a finite domain, where each condition variable $c_i$ takes values in a finite set $\mathcal{V}_i$. For an arbitrary value $a_m \in A$, we define the indicator function:

$$f(\mathbf{v}) = \mathbf{1}[g(\mathbf{v}) = a_m], \quad \mathbf{v} \in \prod_{i=1}^{k} \mathcal{V}_i. \tag{15}$$

Since the domain of $g$ is finite, $f$ can be expressed in disjunctive normal form (DNF):

$$f(\mathbf{v}) = \bigvee_{\mathbf{v} \in \mathcal{C}_{a_m}} \left( \bigwedge_{i=1}^{k} (c_i = v_i) \right), \quad \text{where } \mathcal{C}_{a_m} = \{\mathbf{v} \mid g(\mathbf{v}) = a_m\}. \tag{16}$$

We construct a conditional reasoning DAG $\mathcal{G} = (\mathcal{N}, \mathcal{E})$ such that:

- Each internal node corresponds to a condition variable $c_i$;
- Each edge from a node $c_i$ is labeled by a value $v \in \mathcal{V}_i$;
- Each root-to-leaf path encodes a conjunction $\bigwedge_{i=1}^{k} (c_i = v_i)$ for some $\mathbf{v} \in \mathcal{C}_{a_m}$;
- Each leaf node is labeled with $a_m$.

Formally, for each $\mathbf{v} = (v_1, \ldots, v_k) \in \mathcal{C}_{a_m}$, we construct a path $P_{\mathbf{v}} = (n_0, n_1, \ldots, n_k)$ where:

- $n_0$ is the root node,
- for each $j = 1, \ldots, k$, node $n_j$ is labeled with $c_{\pi(j)}$ for some fixed total order $\pi$ over $[k]$,
- edge $(n_{j-1}, n_j)$ is labeled by $v_{\pi(j)}$,
- the final node $n_k$ connects to a terminal node labeled with $a_m$.

By construction, the union of all such root-to-leaf paths exactly encodes the DNF of $\mathbf{1}[g(\cdot) = a_m]$. □

## B  ANALYSIS OF CLARIFICATION EFFICIENCY.

To analyze the efficiency of our clarification strategy, we note that the worst-case number of clarifications is bounded by the total number of condition variables $k$. However, in practice, each conclusion typically depends on only a small subset of these variables. We denote the average number of conditions along a valid reasoning path as $r \ll k$. Our dynamic traversal algorithm prunes inconsistent branches based on user responses, and selects the most cost-effective clarification at each step. As a result, the expected number of clarification turns is reduced to $O(r)$. Moreover, the DAG structure allows for node sharing across multiple paths, which enables information reuse and further reduces the overall number of clarification turns below $r$ in settings with high condition overlap across paths.

## C  PROMPTS

This section presents the prompts used in our method, including DAG extraction prompt for clarification, conditional path guided data synthesis prompt for DAG construction and evaluation prompt.

## C.1 DAG EXTRACTION PROMPT FOR CLARIFICATION

---
**DAG extraction prompt for clarification**

```
Given a user question and a relevant document that are useful
for answering the question, your task is to:
1.  Based on the passage, decide whether the user question
has multiple conditional answers that are only applicable when
certain user-specific conditions apply.
2.  Then, build a graph (DAG) to represent all possible
conditional reasoning paths.  The node and edge of the DAG
should be json format as follows:
Node format:
{
  "node id":  unique integer ID.
  "node type":  either "Condition" or "Conclusion", "Conclusion"
nodes must be terminal nodes with no outgoing edges.
  "node content":  if the current node is Condition node, the
content should be a clarification question about the conditional
judgement; if the current node is Conclusion node, the content
should be a statement about the final answer to the user's
question.
  "pre node id":  a list of the predecessor nodes of the current
node, if a node has multiple predecessor nodes, the predecessor
nodes are in OR relationship.
}
Edge format:
{
  "from":  the starting node id of the edge, must be a Condition
node.
  "to":  the ending node id of the edge.
  "label":  the label of the edge, should be the answer of the
starting Condition node's clarification question.
}
Your output must contain only two parts:
[nodes] A list of all nodes.  Each node must follow json format
above.  [nodes]
[edges] A list of all edges.  Each edge must follow json format
above.  [edges]
Notice:  Your output must include the above two parts with
complete and properly closed tags.
Now, let's begin:
The user question is:  [query here]
The document is:  [document here]
Output:
```
---

## C.2 CONDITIONAL PATH GUIDED DATA SYNTHESIS PROMPT

---
**Conditional path guided data synthesis prompt**

```
Your task is to extract structured decision problems from
the following policy document.  These problems must meet the
following criteria:
1.  The question has multiple possible answers (not just
yes/no).
2.  The answer depends on two or more user-specific conditions.
3.  Different combinations of these conditions lead to different
answers.
```
---

```
For each decision problem you identify, extract the following
fields:
- "question":  A concise question that summarizes the decision
in the first person.
- "conditions":  A list of all relevant condition checks
in natural language.  These should be simple yes/no-type
evaluations.
- "outputs":  A list of reasoning paths.  Each path should
contain:
  - a combination of condition values (e.g., "A":  "Yes", "B":
"No")
  - the resulting answer
  - a brief natural language explanation of the reasoning
Use the following output format:
[
  {
    "question":  "...",
    "conditions":  ["..."],
    "outputs":
    [
      {
      "combination":  {"...":  "...", "...":  "..."},
      "answer":  "...",
      "reason":  "..."
      }
    ]
  }
]
Notice:  Your output must contain only the list with no other
words!
Now process the following document and extract all such
multi-conditional decision problems.  The document is:
 [document here]
```

## C.3  EVALUATION PROMPT

**Evaluation prompt**

```
Given a question, a candidate answer, and a ground truth answer,
your task is to determine whether the candidate answer is
semantically consistent with the ground truth answer based on
the following criteria:
Semantic Consistency Rules
1.  If the ground truth answer contains a single definite
conclusion, the candidate answer should express the same
conclusion.
2.  The candidate answer must not introduce any conclusions that
contradict the ground truth answer.
Output Format
Your output should consist of two parts:  a reasoning part and
a conclusion part.  The reasoning part should explain your
judgment process.  The conclusion part's content is "yes" if
two answers are considered semantically consistent, otherwise
"no".
The question is:  [question here]
The ground truth answer is:  [ground truth answer here]
The candidate answer is:  [candidate answer here]
```

## D  ALGORITHM OF DYNAMIC TRAVERSAL-BASED CLARIFICATION

Alg. 1 shows the detailed procedure of dynamic traversal-based clarification.

---

**Algorithm 1** Dynamic Traversal-Based Clarification

---

**Require:** DAG $\mathcal{G} = (\mathcal{N}, \mathcal{E})$, Clarifier LLM $\Theta_C$, User Simulator LLM $\Theta_U$ with access to background $S$, known condition set $C_{\text{known}}(q)$
**Ensure:** Final answer $\hat{a}$
 1: Initialize dialogue history $H \leftarrow \emptyset$
 2: Compute candidate set $U$ according to Eq. 1
 3: **while** $U \neq \emptyset$ **do**
 4:     Select $n_i$ according to Eq. 2
 5:     Generate clarification question $q_{n_i} \sim \Theta_C(n_i)$
 6:     Obtain user response $a_{n_i} \sim \Theta_U(q_{n_i}, S)$
 7:     **if** $\exists(n_i, n_j, \nu) \in \mathcal{E}, \nu \equiv a_{n_i}$ **then**
 8:         Record $(q_{n_i}, a_{n_i})$ into $H$
 9:         Continue traversal to $n_j$
10:     **else**
11:         Remove $n_i$ from $U$ and continue with next candidate
12:     **end if**
13: **end while**
14: **return** $\hat{a} \leftarrow \Theta_C(H)$

---

## E  ALGORITHM OF GPS INFERENCE PROCESS

Alg. 2 shows the overall inference pipeline of GPS, which consists of conditional reasoning DAG construction and dynamic traversal-based clarification (Alg. 1).

---

**Algorithm 2** GPS Inference: Graph-guided Proactive Clarification

---

**Require:** User query $q$, retrieved document $d$, Reasoner LLM $\Theta_R$, Clarifier LLM $\Theta_C$, User (or Simulator) $\Theta_U$ with background $S$
**Ensure:** Final answer $\hat{a}$
 1: **// Reasoning stage: DAG construction**
 2: Construct a DAG-extraction prompt $P_{\text{DAG}}(q, d)$ (see Appendix C.1).
 3: Generate a structured DAG description $y \sim \Theta_R(P_{\text{DAG}}(q, d))$.
 4: Parse $y$ into a conditional reasoning DAG $\mathcal{G} = (\mathcal{N}, \mathcal{E}) = \text{PARSE}(y)$.
 5: **// Clarification stage: dynamic traversal (Alg. 1)**
 6: Identify the known condition set $C_{\text{known}}(q) \leftarrow \Theta_C(q, d, \mathcal{G})$.
 7: $\hat{a} \leftarrow \text{DYNAMICTRAVERSALCLARIFICATION}(\mathcal{G}, \Theta_C, \Theta_U, C_{\text{known}}(q), S)$
 8: **return** $\hat{a}$

---

## F  ALGORITHM OF THE REASONER TRAINING PROCESS

Alg. 3 summarizes the clarification-oriented reinforcement learning procedure used to train the Reasoner LLM $\Theta_R$ with hybrid rewards over synthetic conditional-path data.

## G  CASE STUDY

### G.1  QUALITATIVE COMPARISON BETWEEN GPS AND BASELINE METHODS

Figure 4 illustrates the different clarification processes adopted by GPS and ProCoT on the same underspecified query from the **Synthetic** dataset. GPS successfully identifies the correct conditional

---

**Algorithm 3** Clarification-Oriented RL for DAG Extraction

---

**Require:** Document collection $\mathcal{D}$, data synthesis module SYNTH, initial Reasoner LLM $\Theta_R^{(0)}$, Clarifier LLM $\Theta_C$, User Simulator $\Theta_U$, RL iterations $T$.

**Ensure:** Trained Reasoner LLM $\Theta_R^{(T)}$.

1: **// Conditional-path guided data synthesis**
2: $\mathcal{S} \leftarrow \text{SYNTH}(\mathcal{D})$
3: **for** $t = 1$ **to** $T$ **do**
4:     Sample a minibatch $\mathcal{B} \subset \mathcal{S}$
5:     **for each** $(q, d, a, C_{\text{miss}}) \in \mathcal{B}$ **do**
6:         **// DAG extraction by current Reasoner**
7:         $\mathcal{G} = (\mathcal{N}, \mathcal{E}) \sim \Theta_R^{(t-1)}(q, d)$
8:         **// Simulated clarification and answer prediction**
9:         $\hat{a}, T_{\text{clar}} \leftarrow \text{DYNAMICTRAVERSALCLARIFICATION}\big(\mathcal{G}, \Theta_C, \Theta_U, C_{\text{known}}(q), S\big)$
10:        **// Hybrid reward computation**
11:        Compute total reward $R$ as in Eq. 13
12:        Store $(q, d, \mathcal{G}, R)$ for RL update
13:     **end for**
14:     **// RL update of Reasoner**
15:     $\Theta_R^{(t)} \leftarrow \text{RLUPDATE}(\Theta_R^{(t-1)}, \{(q, d, \mathcal{G}, R)\}_{(q,d,\cdot) \in \mathcal{B}})$
16: **end for**
17: **return** $\Theta_R^{(T)}$

---

rules from the document (highlighted in red) and constructs a conditional reasoning DAG to guide the clarification process through traversal, ultimately leading to the correct answer. In contrast, Pro-CoT is distracted by irrelevant information in the document (highlighted in orange), asks unrelated clarification question, and consequently derives an incorrect answer.

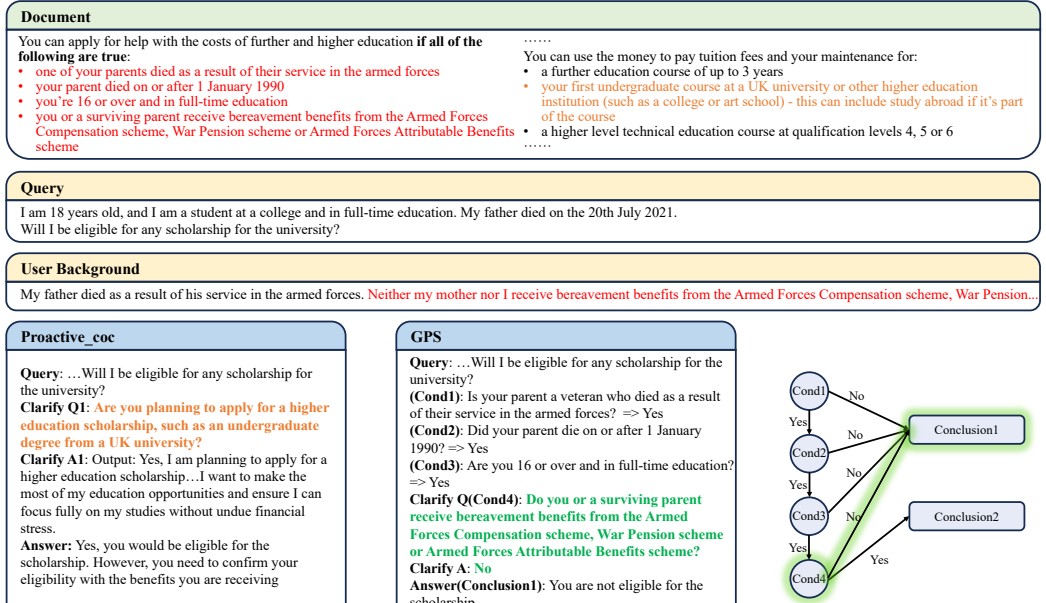

Figure 4: Comparison of GPS and ProCoT on an underspecified query from the **Synthetic** dataset. GPS extracts the correct conditional rules and uses the resulting DAG to ask the necessary clarification and reach the correct answer, whereas ProCoT focuses on irrelevant details and asks an unrelated question, leading to an incorrect conclusion.

Figure 5 illustrates that GPS provides a more reliable clarification process than UoT. UoT selects clarification questions based on uncertainty signals but lacks an explicit representation of the full decision structure. As a result, it checks only the first two conditions and stops once uncertainty appears reduced, which causes it to miss a decisive eligibility factor and produce an incorrect conclusion. In contrast, GPS constructs the conditional reasoning DAG, queries all necessary conditions through dynamic traversal, and therefore arrives at the correct final answer.

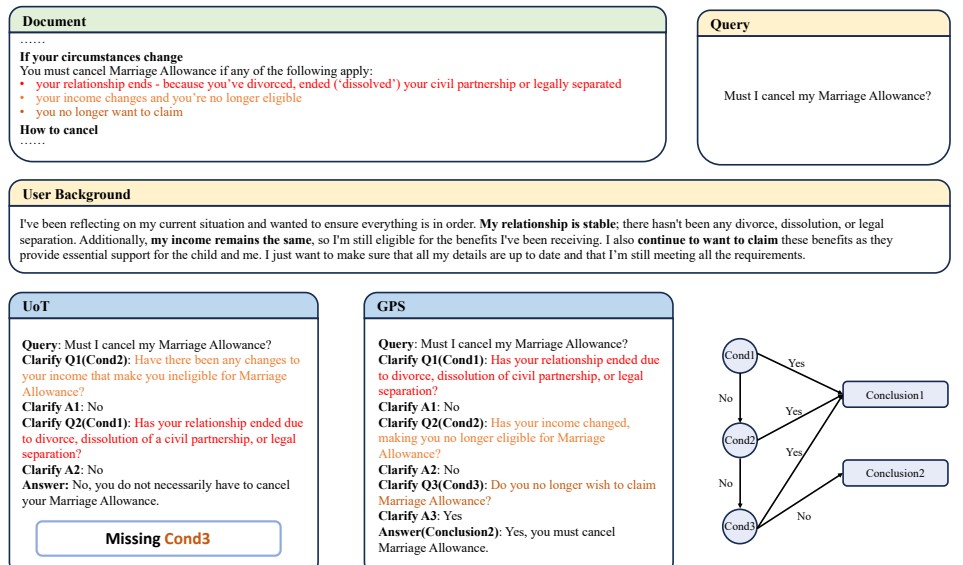

Figure 5: GPS provides a more reliable clarification process than UoT. UoT queries only part of the relevant conditions and terminates prematurely, which causes it to miss a decisive eligibility factor and produce an incorrect conclusion. GPS constructs the full conditional reasoning DAG, queries all necessary conditions through structured traversal, and therefore arrives at the correct final answer.

### G.2    CAPABILITY OF GPS IN MODELING NESTED CONDITIONAL LOGIC

Figure 6 demonstrates that the underlying rule structure is not a flat sequence of conditions but a genuinely nested logical hierarchy. The eligibility decision depends on multiple interacting sub-rules: an initial branch based on registration date, a second layer involving engine standard, and a third layer contingent on the presence of a filter. In parallel, a separate subtree handles converted or re-engined vehicles, further routing to different authorities depending on subsequent conditions. These rule blocks depend on one another in a layered manner, where the outcome of one condition determines which deeper sub-rule becomes applicable—a defining characteristic of nested logic.

Under our framework, such hierarchical dependencies map cleanly into a DAG. Conjunctive dependencies appear as chained edges, disjunctive alternatives as branching nodes, and intermediate outcomes naturally serve as parent nodes for deeper conditional layers. As stated in Proposition 1, any finite conditional rule system with nested structure can be transformed into such a DAG without loss of logical fidelity. The figure illustrates this concretely: GPS successfully captures all nested branches in a structurally precise DAG, confirming that the method faithfully models and traverses multi-level logical hierarchies rather than only simple condition–conclusion patterns.

### G.3    IMPACT OF STRUCTURAL QUALITY REWARDS ON DAG CONSTRUCTION

Figure 7 illustrates how the proposed structural quality reward $r_\eta$ distinguishes between well-structured and poorly-structured clarification DAGs. The left DAG generated by GPS forms a clean hierarchical decision structure: each clarification introduces meaningful discrimination, branches do not recombine, and each split contributes directly to narrowing the final conclusions. As a result, its split entropy is fully converted into leaf-level discriminative power, yielding $r_\eta = 1$.

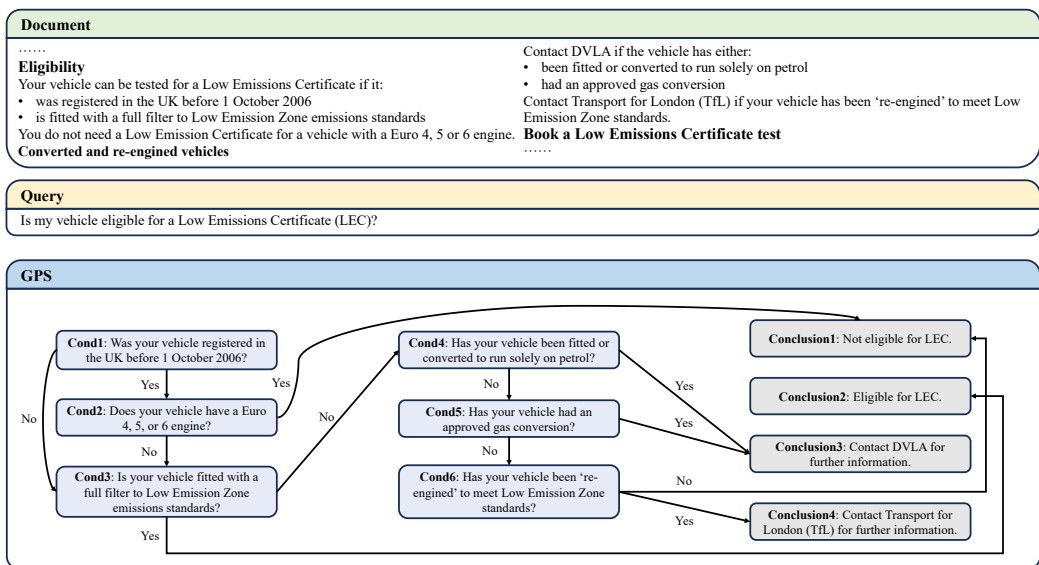

Figure 6: GPS successfully models a multi-layer nested logical hierarchy: conjunctive chains, disjunctive branches, and deeper sub-rules are all represented in a unified DAG structure.

In contrast, the right DAG generated by backbone model exhibits redundant branching: several clarifications produce splits that later merge, creating patterns where injected uncertainty does not contribute to distinguishing final leaves. This causes the graph-level split entropy $H_{\mathrm{graph}}$ to increase while the leaf entropy $H_{\mathrm{leaf}}$ remains low, yielding a substantially reduced score of $r_\eta = 0.46$.

This case demonstrates that the structural quality reward effectively penalizes DAGs whose intermediate clarifications do not help refine the final conclusion space, and correspondingly encourages models to produce non-redundant clarification structures.

## H  ILLUSTRATION OF THE OVERALL GPS REASONING AND CLARIFICATION WORKFLOW

Figure 8 provides a concise end-to-end illustration of the GPS framework. Given a user query and its associated document, the Reasoner first extracts all condition-dependent rules and generates a conditional reasoning DAG, where internal nodes represent clarification conditions and leaf nodes represent possible conclusions. Based on this DAG, the Clarifier interacts with the user in a traversal manner, issuing only the condition queries necessary to eliminate incompatible branches. As user responses progressively constrain the DAG, the traversal converges to a unique conclusion, from which the final answer is produced.

This example illustrates how GPS combines document-grounded rule extraction with adaptive clarification to resolve underspecified query effectively.

## I  IMPLEMENTATION DETAILS

The experiments are conducted on a machine equipped with 8 NVIDIA A800 GPUs. For GRPO, we apply LoRA and set the rank of LoRA to 64. The training epoch is set to 1, the batch size is set to 32 and the learning rate is set to 3e-6. The hyperparameter $\alpha$ in the hybrid reward is set to 0.5.

## J  ANALYSIS OF PERFORMANCE ON TWO TYPES OF QUERIES

Table 3 reports the SR (%) of different methods on ConditionalQA and ShARC, separately for underspecified and well-specified queries. On ConditionalQA, GPS consistently outperforms all base-

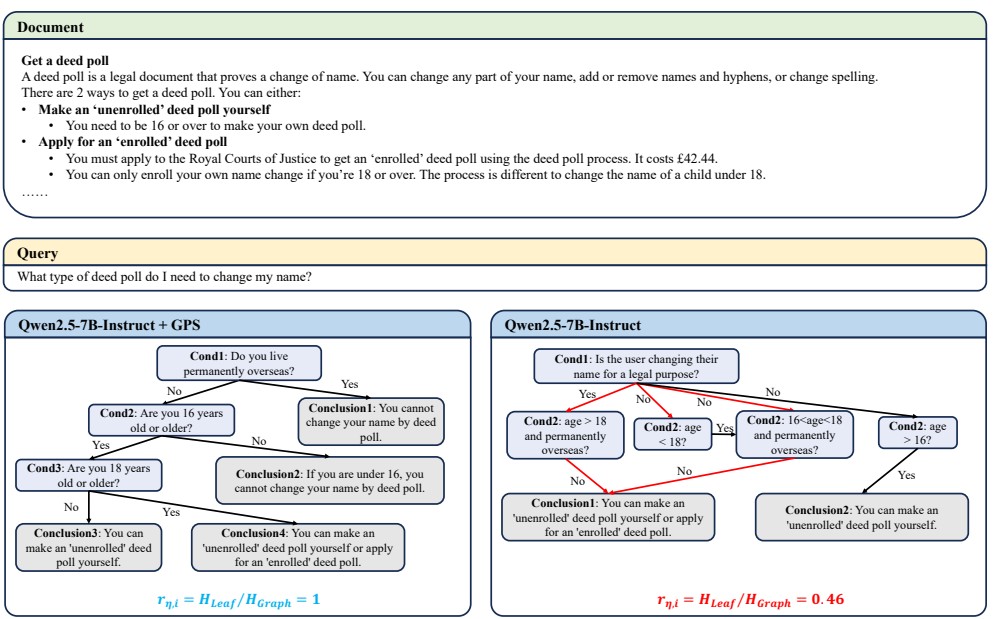

Figure 7: Comparison of structural quality between two clarification DAGs. The GPS-generated structure (left) forms clean, monotonic decision refinement and achieves $r_\eta = 1$. The baseline (right) contains redundant branching and split–merge patterns, which inflate $H_{\mathrm{graph}}$ without increasing $H_{\mathrm{leaf}}$, resulting in a low $r_\eta = 0.46$. The structural quality reward explicitly captures this efficiency gap and drives learning toward well-structured clarification DAGs.

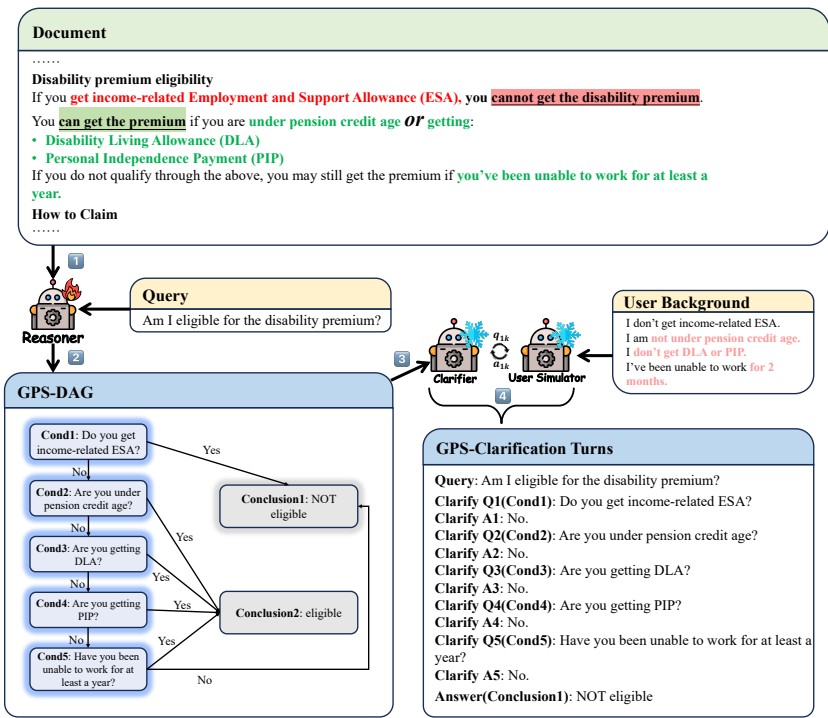

Figure 8: Overview of the GPS workflow. The Reasoner first extracts condition-dependent rules from the query–document pair and produces a conditional reasoning DAG. The Clarifier then performs multi-turn clarification by traversing the DAG, pruning incompatible branches based on user responses, and converging to a valid conclusion.

lines, achieving the highest SR in both well-specified (72.7) and underspecified (81.1), demonstrating the effectiveness of DAG-guided clarification. ProCoT also performs competitively on underspecified queries (69.8), surpassing Base Method and Clarify-DPO. In contrast, BED-LLM shows consistently poor performance, especially in well-specified queries (46.6). On ShARC, Clarify-DPO achieves the best performance on underspecified queries (91.4), while GPS remains strong and balanced across both settings (71.6/80.0). Interestingly, Base Method and ProCoT collapse on ShARC underspecified queries (32.3), suggesting limited cross-domain generalization. Overall, these results highlight the robustness of GPS in identifying and resolving ambiguity.

Table 3: Success Rate (SR, %) on well-specified vs. underspecified queries across ConditionalQA and ShARC with the LLaMA backbone. **Bold** denotes the best result and underline the second-best.

| Method | ConditionalQA | | ShARC | |
|---|---|---|---|---|
| | Well-specified | Underspecified | Well-specified | Underspecified |
| Base Method | 63.6 | 60.4 | **80.9** | 32.3 |
| UoT | 64.8 | 32.9 | 69.1 | 67.7 |
| Clarify-DPO | 68.2 | 53.9 | 74.1 | **91.4** |
| ProCoT | 65.3 | 69.8 | 75.2 | 32.3 |
| BED-LLM | 46.6 | 40.5 | 64.4 | 63.6 |
| GPS | **72.7** | **81.1** | 71.6 | 80.0 |

## K  DATASET DETAILS

We first construct a Synthetic dataset consisting of 2575 underspecified queries based on documents from ConditionalQA. These samples are split into training and testing sets using 8:2 ratio, with 2060 samples used for training and the remainder for testing. For overall training, we combine the Synthetic training split with the underspecified queries of the ConditionalQA training set, resulting in a total of 3250 training samples. For evaluation, we use three evaluation datasets: the Synthetic test split, the ConditionalQA test set, and the ShARC test set. We present the detailed size of our training dataset and three evaluation benchmark datasets in Table 4.

Table 4: Dataset Statistics.

| Sources | Underspecified | Well-specified | Total |
|---|---|---|---|
| **Training data** | | | |
| Synthetic | 3250 | 0 | 3250 |
| **In-domain test data** | | | |
| Synthetic | 515 | 0 | 515 |
| ConditionalQA | 53 | 176 | 229 |
| *In-domain Total* | 568 | 176 | 744 |
| **Out-of-domain test data** | | | |
| ShARC | 675 | 675 | 1350 |

## L  THE USE OF LARGE LANGUAGE MODELS (LLMS)

In this work, Large Language Models (LLMs) are mainly used as auxiliary tools rather than core components of the proposed method. Specifically, we leverage LLMs for two purposes: (i) grammar checking and language polishing of academic writing; and (ii) providing suggestions for code

debugging, particularly in identifying possible causes of error messages and offering potential fixes. These uses of LLMs help streamline the writing and coding workflow, but they do not influence the methodological design or experimental results of this paper.

