# OpenReview forum: "GPS: Graph-guided Proactive Information Seeking in Large Language Models"
_ICLR.cc/2026/Conference — ICLR 2026 Poster_

### Official Review · Reviewer_behg · 2025-10-23

**Soundness:** 1
**Presentation:** 2
**Contribution:** 1
**Rating:** 2
**Confidence:** 4

**Summary:**

The GPS (Graph Guided Proactive Information Seeking) framework proposed in this paper addresses the issue of active clarification in RAG systems when LLM processes queries with insufficient information. It innovatively introduces a conditional inference structure based on directed acyclic graphs (DAGs) and combines conditional path guided data synthesis with clarification oriented reinforcement learning, effectively balancing the effectiveness and interaction efficiency of clarification.

**Strengths:**

1. The design of DAG conditional reasoning structure combines logic and efficiency, breaking through the limitations of traditional prompting methods that rely on LLM spontaneous reasoning, and providing structured logical support for active clarification.

2. The conditional path guided data synthesis method not only optimizes the logical correctness of DAG, but also suppresses redundant interactions and structural redundancy, achieving multi-objective collaborative optimization.

**Weaknesses:**

1. The specific implementation logic for extracting DAG is not clear, such as "how to parse conditional variables and logical relationships from unstructured documents" and "when there are fuzzy rules in the document.

2. Suggest adding parameter sensitivity experiments to clarify the optimal parameter selection strategy; Fully derive the calculation process of structural quality rewards and verify its rationality through examples.

3. In the relevant work section, there was no in-depth comparison of the essential differences between GPS and existing models in "structured inference targets" In addition, many SOTA methods have already studied knowledge graphs and their logic. What is the difference between this article and them?

**Questions:**

1. Why was the method in this article not compared with the GraphRAG series models?

2. The second stage of this method uses models around 7B, which is too small compared to the deepseek used in the first stage. Should we use LLMs of the same scale? Existing mainstream methods also tend to use larger models

---

> ### Author Response · Authors · 2025-11-23
>
> We sincerely appreciate your comments, suggestions, and every effort spent on reviewing our work. Here we attempt to address all your remaining concerns. In the following, we quote your comments and then give our detailed response point-by-point.
>
> > **W1: The specific implementation logic for extracting DAG is not clear, such as "how to parse conditional variables and logical relationships from unstructured documents" and "when there are fuzzy rules in the document".**
>
> Thank you for the helpful suggestion. We briefly summarize here how we perform DAG extraction with the Reasoner. As described in our paper, we leverage the strong instruction-following capability of large language models and design a dedicated prompt for the DAG extraction. The prompt is provided in Appendix G.1. Below we outline the core instructions given to the Reasoner:
>
> **1. Detect conditional structure.** Determine whether the document contains multiple condition-dependent outcomes for the given query.
>
> **2. Construct conditional reasoning DAG.** Identify the relevant condition variables and organize the different condition–conclusion mappings into a DAG. Following the design described in Section 4.1, the prompt explicitly specifies the JSON node–edge schema to ensure that the extracted DAG adheres to the intended structure.
>
> We further strengthen the DAG extraction ability of the Reasoner through reinforcement learning. In our updated manuscript, we provide pseudo-code for the GPS inference procedure in Appendix D, the RL training algorithm for the Reasoner in Appendix E, and we include an illustrative example of the overall GPS information-seeking workflow in Appendix I.
>
> Regarding fuzzy rules, our datasets mainly contain crisp eligibility conditions, but we claim that our DAG structure can accommodate fuzzy statements by incorporating them into the conclusion text rather than treating them as branching conditions.
>
> We additionally evaluate GPS on **Abg-CoQA**[5], an open-domain conversational QA dataset explicitly designed for **ambiguity detection and clarification in natural, fuzzy contexts**. Abg-CoQA covers ambiguity types such as coreference, event references, temporal uncertainty, and answer-type ambiguity, making it a suitable benchmark for evaluating transfer to **less structured, non-rule-based settings**. The results are shown below:
>
> | Method               | SR ↑ | WCT ↓| F1 ↑ |
> |----------------------|------|------|-------|
> | Base Method          |   46.9  | 5.31 |   0.0   |
> | ProCoT               | 43.0 | 5.78 | 17.6  |
> | Adaptive Clarify-DPO | 55.4 | 4.46 | 0.0   |
> | GPS (ours)           | **57.1** | **4.45** | **53.7**  |
>
> The results show that GPS demonstrates clear improvements over baseline methods across all metrics, indicating that **its structural reasoning ability transfers effectively to fuzzy, conversational environments**, even when conditional relationships are not explicitly rule-governed.
>
> [1] Guo M, Zhang M, Reddy S, et al. Abg-CoQA: Clarifying Ambiguity in Conversational Question Answering[C]. AKBC, 2021.

---

> ### Author Response · Authors · 2025-11-23
>
> > **W2: Suggest adding parameter sensitivity experiments to clarify the optimal parameter selection strategy; Fully derive the calculation process of structural quality rewards and verify its rationality through examples.**
>
> Thank you for the helpful suggestion. For the first question, we conduct the parameter sensitivity experiments on both Synthetic and ConditionalQA datasets. The results are shown below:
>
> **Parameter Sensitivity on Synthetic (Qwen2.5-7B-Instruct)**
>
> | α  | SR ↑    | WCT ↓   | F1 ↑    |
> |--------|--------|--------|--------|
> | 0.25    |   54.4    |   5.30    |   93.7    |
> | 0.5    |   60.2    |   4.59    |   96.4    |
> | 0.75    |   58.2    |   4.88    |   94.5    |
> | 1.0    |   55.3    |   5.15    |   95.7    |
>
> **Parameter Sensitivity on ConditionalQA (Qwen2.5-7B-Instruct)**
> | α  | SR ↑    | WCT ↓   | F1 ↑    |
> |--------|--------|--------|--------|
> | 0.25    |   72.5    |   3.11    |   27.1    |
> | 0.5    |   73.4    |   2.91    |   36.7    |
> | 0.75    |   72.4    |   3.09    |   28.8    |
> | 1.0    |   75.1    |   2.66    |   13.9    |
>
> We choose α = 0.5 because it provides the **best balance between correctness and efficiency**, yielding the highest overall performance across SR, WCT, and F1 on both datasets.
>
> For the second question, in our updated manuscript, we provide a more complete derivation of the structural quality reward in section 4.3.2 and provide an illustrative example in the Appendix H.3 to demonstrate why the reward assigns higher scores to well-structured DAGs and lower scores to incomplete or inconsistent ones.

---

> ### Author Response · Authors · 2025-11-23
>
> > **W3: In the relevant work section, there was no in-depth comparison of the essential differences between GPS and existing models in "structured inference targets". In addition, many SOTA methods have already studied knowledge graphs and their logic. What is the difference between this article and them?**
>
> Thank you for the helpful suggestion. We provide the comparison between GPS and related work as follows:
>
> **Clarification-oriented methods:** Existing clarification-oriented methods such as UoT, BED-LLM, and Clarify-DPO treat clarification as an open-ended dialogue generation problem. Because their structured targets are limited to linear dialogue trajectories or uncertainty scores, these methods often struggle under complex rule contexts, **leading to missed necessary conditions and incomplete clarification paths**, as demonstrated in our updated Sec. 5 and Appendix H.
>
> In contrast, **GPS uses a conditional reasoning DAG as the structured inference target**, which is **theoretically complete** and supports **dynamic pruning** for efficient clarification. With this explicit logical structure, GPS **captures all relevant branches and produces correct, minimal clarification paths**, even in documents with rich conditional dependencies.
>
> **Graph-based and KG-based methods:** These methods indeed operate on structured objects, such as **problem-specific thought graphs** [2], **logical query graphs over static KG** [3,4], and **static KGs** [5]. However, the structured targets underlying these methods differ fundamentally from ours: they primarily encode entities, relations, or intermediate reasoning steps for **well-specified queries**, with the objective of improving multi-hop reasoning performance or the faithfulness of logical inference over the structures.
>
> In contrast, **GPS is explicitly designed for underspecified queries**, where key user-specific conditions are missing and must be elicited. GPS therefore **dynamically constructs a query-specific conditional reasoning DAG from unstructured documents**. This makes our DAG a different kind of structured inference target. Our DAGs are tailored to representing rule dependencies and support efficient pruning for resolving underspecified queries.
>
> [2] Besta M, Blach N, Kubicek A, et al. Graph of Thoughts: Solving Elaborate Problems with Large Language Models[C]. AAAI, 2024.
>
> [3] Ren H, Hu W, Leskovec J. Query2box: Reasoning over Knowledge Graphs in Vector Space using Box Embeddings[C]. ICLR, 2020.
>
> [4] Xia T, Ding L, Wan G, et al. Improving Complex Reasoning over Knowledge Graph with Logic-Aware Curriculum Tuning[C]. AAAI, 2025.
>
> [5] Gutiérrez B J, Shu Y, Gu Y, et al. HippoRAG: Neurobiologically Inspired Long-Term Memory for Large Language Models[C]. NeurIPS, 2024.

---

> ### Author Response · Authors · 2025-11-23
>
> > **Q1: Why was the method in this article not compared with the GraphRAG series models?**
>
> To the best of our knowledge and according to survey [6], GraphRAG methods focus on building entity–relation or summary graphs over document corpora to support multi-hop retrieval and answer generation for open-domain or complex QA. The main goal is to improve **what to retrieve and how to aggregate the retrieved knowledge** for answering a complex well-specified question.
>
> In contrast, our work focuses on **underspecified questions**, aiming to enhance the model's ability to **detect such ambiguity** and to perform **active information seeking**. Therefore, we view GraphRAG methods and our method as **orthogonal**: GraphRAG improves the quality of the retrieved knowledge, while our method resolves user-specific ambiguity given the retrieved knowledge.
>
> > **Q2: The second stage of this method uses models around 7B, which is too small compared to the deepseek used in the first stage. Should we use LLMs of the same scale? Existing mainstream methods also tend to use larger models.**
>
> A growing line of work such as model compression[7,8] and knowledge distillation[9,10] aims to improve the reasoning capabilities of compact models (7B-13B) that are practical for real-world deployment. These mid-sized models are easier to deploy under latency or memory constraints and compatible with on-premise or privacy-sensitive environments. Similarly, Clarify-DPO, the main training-based baseline in our paper, is also trained on 7B–8B backbones, which reflects standard practice in the proactive clarification literature. Nonetheless, we believe that exploring the GPS framework with larger-scale LLMs under more abundant compute resources is an interesting direction for future research.
>
> [6] Zhang Q, Chen S, Bei Y, et al. A Survey of Graph Retrieval-Augmented Generation for Customized Large Language Models[R]. arXiv, 2025.
>
> [7] Sun M, Liu Z, Bair A, et al. A Simple and Effective Pruning Approach for Large Language Models[C]. ICLR, 2024.
>
> [8] Ma X, Fang G, Wang X. LLM-Pruner: On the Structural Pruning of Large Language Models[C]. NeurIPS, 2023.
>
> [9]Gu Y, Dong L, Wei F, et al. MiniLLM: Knowledge Distillation of Large Language Models[C]. ICLR, 2024.
>
> [10] Liu J, Zhang C, Guo J, et al. DDK: Distilling Domain Knowledge for Efficient Large Language Models[C]. NeurIPS, 2024.

---

> > ### Author Response · Authors · 2025-11-27
> >
> > Dear Reviewer,
> >
> > I hope this message finds you well. We extend our gratitude once more for your valuable and insightful comments!
> >
> > We have provided careful and detailed responses to all your questions. It would be greatly appreciated if you could kindly let us know whether we have answered all your questions. Please also kindly let us know if you have any further questions, and we would like to try our best to resolve them before the deadline.
> >
> > Best regards, Authors of the paper 22850

---

### Official Review · Reviewer_1Jai · 2025-10-29

**Soundness:** 3
**Presentation:** 2
**Contribution:** 2
**Rating:** 6
**Confidence:** 3

**Summary:**

The paper introduces GPS, a two-stage framework that enables large language models in retrieval-augmented generation systems to proactively clarify underspecified user queries by modeling conditional logic in retrieved documents as a directed acyclic graph. In the reasoning stage, a Reasoner LLM constructs a logically complete graph that captures the AND and OR relationships among condition variables and possible answers. In the clarification stage, a Clarifier LLM dynamically traverses the graph, selecting questions from a candidate set of nodes based on their expected remaining depth, and prunes inconsistent paths according to user responses to achieve efficient clarification with only a few turns.

To train the Reasoner, the authors propose conditional path-guided synthesis that augments ConditionalQA by generating underspecified queries with multiple path assignments and filtering them through a Verifier LLM to ensure consistency. They further apply reinforcement learning with a hybrid reward combining accuracy, efficiency, and structural quality to enhance reasoning precision and interaction efficiency.

**Strengths:**

- Proposition 1 rigorously proves logical completeness via DNF encoding, with every root-to-leaf path corresponding to a conjunction.
- Conditional path-guided generation from documents produces underspecified queries with explicit missing conditions and corresponding reasoning paths, each consisting of variable–answer pairs. These queries are filtered based on necessity—retained only if a Verifier model can answer correctly when given the full conditions but fails when any are masked. This process yields high-quality augmentations of ConditionalQA’s limited underspecified samples, expanding the dataset significantly without requiring human annotation.

**Weaknesses:**

- Filtering retains samples only when the Verifier LLM predicts the correct answer under full conditions but fails under partial ones. However, this approach discards nuanced cases involving partial ambiguity resolution and inherits the biases of the Verifier model, such as inconsistencies observed in DeepSeek-R1.
- There is no inter-annotator agreement or human validation for the roughly 75.5% of samples discarded during ConditionalQA augmentation.

**Questions:**

- How often does the Reasoner generate invalid directed acyclic graphs—for instance, containing cycles, incomplete edge coverage, or mismatched condition variable domains—and what fallback mechanism is applied during traversal if graph parsing fails or if the candidate set becomes empty prematurely?
- Regarding the efficiency reward, what specific value of α was used, and how does the model’s performance degrade when user responses deviate from the simulator assumptions, such as providing evasive answers or multi-valued inputs outside the defined variable domains?

---

> ### Author Response · Authors · 2025-11-23
>
> We sincerely appreciate your comments, suggestions, and every effort spent on reviewing our work. Here we attempt to address all your remaining concerns. In the following, we quote your comments and then give our detailed response point-by-point.
>
> > **W1: Filtering retains samples only when the Verifier LLM predicts the correct answer under full conditions but fails under partial ones. However, this approach discards nuanced cases involving partial ambiguity resolution and inherits the biases of the Verifier model, such as inconsistencies observed in DeepSeek-R1.**
>
> Thank you for this careful observation and insightful comment. Our filtering strategy is deliberately conservative: we only retain cases where the Verifier is correct under full conditions but fails under partial ones, because these provide a *high-precision* signal that the missing condition is truly essential. In contrast, partially resolved cases where the Verifier answers correctly under some partial subsets but inconsistently under others can be noisy and unreliable without human annotation. For example, in a document where eligibility depends jointly on *age* and *residency*, a Verifier may guess the correct answer when only *age* is provided but fail when only *residency* is provided, even though both conditions are required. Treating such fluctuations as reliable labels would misidentify the true source of ambiguity and introduce supervision noise.
>
> Regarding the concern about inheriting biases from the Verifier, we emphasize that we mitigate this issue by performing **majority voting**, a technique shown to substantially improve reasoning stability and accuracy in prior work[1]. This reduces the impact of individual inconsistent predictions, including those observed in DeepSeek-R1, while preserving the high-precision nature of the filtering rule.
>
> We hope to clarify that the filtering mechanism is a pragmatic way to obtain clean supervision under limited annotation budget. Although it excludes some nuanced partial-resolution cases, these cases are intrinsically noisy, and excluding them does not undermine the validity of our empirical conclusions.
>
> [1] Wang X, Wei J, Schuurmans D, et al. Self-Consistency Improves Chain of Thought Reasoning in Language Models[C]. ICLR, 2023.
>
> > **W2: There is no inter-annotator agreement or human validation for the roughly 75.5% of samples discarded during ConditionalQA augmentation.**
>
> We would like to clarify how we use the ConditionalQA dataset. We do not introduce any new annotations: our identification of **underspecified questions** directly reuses ConditionalQA's official distinction between **conditional-answer** and **non-conditional** samples. In ConditionalQA, a question is labeled as having conditional answers precisely when the final answer **depends on missing conditions**, which aligns exactly with our definition of underspecification. Thus, the "discarded 75.5%" simply refers to ConditionalQA's **non-conditional** samples, not to any samples we re-annotated. ConditionalQA already reports strong human validation for conditional-answer annotation (two-stage annotation with expert adjudication), and our pipeline inherits this agreement without adding any extra labeling noise.
>
> [2] Sun H, Cohen W, Salakhutdinov R, et al. ConditionalQA: A Complex Reading Comprehension Dataset with Conditional Answers[C]. ACL, 2022.

---

> ### Author Response · Authors · 2025-11-23
>
> > **Q1: How often does the Reasoner generate invalid directed acyclic graphs—for instance, containing cycles, incomplete edge coverage, or mismatched condition variable domains—and what fallback mechanism is applied during traversal if graph parsing fails or if the candidate set becomes empty prematurely?**
>
> Thank you for the insightful question. We fully agree with your categorization of error types, which is consistent with the error types we observe in our experiments, and we indeed incorporate a fallback mechanism to handle such cases. Specifically, we adopt self-reflection mechanism [1]: whenever the generated DAG contains structural errors (e.g., cycles or incomplete node–edge specifications) or when traversal encounters a mismatched condition value, we feed the error message back to the LLM and prompt it to reassess whether its previous graph output is flawed and to regenerate a corrected DAG accordingly. The reflection prompt is provided in Appendix G.
>
> We compute the frequency of three error types, i.e., **cyclic graphs**, **incomplete nodes/edges**, and **traversal failures** for Qwen2.5-7B-Instruct before and after training with GPS on Synthetic dataset. In the table below, each entry is presented in the form "a/b", where the left value corresponds to the frequency before reflection and the right value corresponds to the frequency after reflection.
>
> **Error Type Statistics Before and After GPS Training**
>
> | Model Variant                    | Cyclic Graphs | Incomplete Nodes/Edges | Traversal Failure |
> |----------------------------------|---------------|-------------------------|--------------------|
> | Qwen2.5-7B-Instruct              |       2/0       |            38/11            |         6/2          |
> | Qwen2.5-7B-Instruct + GPS        |       0/0       |            3/0            |         0/0          |
>
> Overall, the error statistics illustrate that GPS produces significantly more accurate and structurally consistent DAGs.
>
> [3] Shinn N, Cassano F, Gopinath A, et al. Reflexion: Language Agents with Verbal Reinforcement Learning[C]. NeurIPS, 2023.
>
> > **Q2.1: Regarding the efficiency reward, what specific value of α was used?**
>
> We set α=0.5 in the efficiency reward at our paper. We also conduct a parameter sensitive experiment to explain the strategy we choose this value. The results are shown below:
>
> **Parameter Sensitivity on Synthetic (Qwen2.5-7B-Instruct)**
>
> | α  | SR ↑    | WCT ↓   | F1 ↑    |
> |--------|--------|--------|--------|
> | 0.25    |   54.4    |   5.30    |   93.7    |
> | 0.5    |   60.2    |   4.59    |   96.4    |
> | 0.75    |   58.2    |   4.88    |   94.5    |
> | 1.0    |   55.3    |   5.15    |   95.7    |
>
> **Parameter Sensitivity on ConditionalQA (Qwen2.5-7B-Instruct)**
> | α  | SR ↑    | WCT ↓   | F1 ↑    |
> |--------|--------|--------|--------|
> | 0.25    |   72.5    |   3.11    |   27.1    |
> | 0.5    |   73.4    |   2.91    |   36.7    |
> | 0.75    |   72.4    |   3.09    |   28.8    |
> | 1.0    |   75.1    |   2.66    |   13.9    |
>
> We choose α = 0.5 because it provides the **best balance between correctness and efficiency**, yielding the highest overall performance across SR, WCT, and F1 on both datasets.

---

> ### Author Response · Authors · 2025-11-23
>
> > **Q2.2: How does the model's performance degrade when user responses deviate from the simulator assumptions, such as providing evasive answers or multi-valued inputs outside the defined variable domains?**
>
> When user responses deviate from the simulator assumptions, the impact depends on whether the affected condition lies on a **critical logical branch** of the DAG. If the mismatched condition corresponds to an essential component of the reasoning chain (e.g., a conjunctive requirement), the model is likely to follow an incorrect branch, which can lead to degraded answer accuracy.
>
> To quantify this effect, we conduct an additional experiment on Synthetic dataset based on Qwen2.5-7B-Instruct backbone: **for each condition that the Clarifier successfully matches to a branch, we perturb the user response with 50% probability**, forcing it no longer matches the subsequent branch. We compare performance under normal user responses and perturbed responses. The metrics include **Success Rate (SR)** and **Weighted Clarification Turns (WCT)**. The results shown below:
>
> **Impact of Perturbed User Responses on SR and WCT**
> | Condition Type      | SR     | WCT    |
> |---------------------|--------|--------|
> | Normal Responses    |   60.2    |   4.59    |
> | Perturbed Responses|   34.8    |   7.05    |
>
> These results show that perturbing user inputs that break key condition matches substantially degrades both accuracy and efficiency, confirming that GPS relies on condition values consistent with the user context along critical reasoning branches.

---

> > ### Author Response · Authors · 2025-11-27
> >
> > Dear Reviewer,
> >
> > I hope this message finds you well. We extend our gratitude once more for your valuable and insightful comments!
> >
> > We have provided careful and detailed responses to all your questions. It would be greatly appreciated if you could kindly let us know whether we have answered all your questions. Please also kindly let us know if you have any further questions, and we would like to try our best to resolve them before the deadline.
> >
> > Best regards, Authors of the paper 22850

---

### Official Review · Reviewer_yNPb · 2025-10-30

**Soundness:** 3
**Presentation:** 2
**Contribution:** 3
**Rating:** 6
**Confidence:** 4

**Summary:**

This paper tackles ambiguity in RAG by teaching LLMs to proactively ask better clarifying questions. It proposes GPS, a two-stage approach: first, represent the retrieved knowledge with a Directed Acyclic Graph so the model can reason over conditional rules in a logically complete way; second, traverse and prune that graph interactively based on user answers to keep clarification efficient. To make this practical, the authors generate training data that reflect conditional paths and further fine-tune with a clarification-oriented RL objective balancing effectiveness and efficiency.

**Strengths:**

1. Rule-structured reasoning: Modeling conditional rules as a graph is a clean, principled way to surface ambiguity and drive targeted clarification, rather than ad-hoc question asking.

2. Theory with practical bite: The logical completeness guarantee plus average-case O(r) clarification complexity gives both soundness and efficiency.

3. End-to-end system design: A coherent pipeline—conditional-path data synthesis, clarification-oriented RL with a hybrid (accuracy/efficiency) reward, and dynamic traversal—aligns training with the actual interaction objective.

**Weaknesses:**

1. Baseline fairness (Clarify-DPO): The original Clarify-DPO does not has the RAG part. It’s unclear whether Clarify-DPO had access to retrieved documents (true RAG) or only engaged in Q&A without retrieval. If the latter, the comparison is unfair; if the former, the paper should specify how evidence was integrated to ensure parity.

2. Training data parity: GPS is trained on ConditionalQA (same policy domain as ShARC). Were baselines also trained on exactly the same splits and sources?

3. Efficiency evidence gap: The claim of higher efficiency isn’t supported by the experiments. There is no comparison in the aspect of efficiency with the baselines.

4. Domain generalization: Evaluations center on rule-heavy policy/regulation datasets; it’s unclear how GPS transfers to domains where rules are fuzzier (open-ended QA, multi-hop encyclopedic tasks) or where conditional structures are incomplete/noisy.

**Questions:**

If the retrieved document is incomplete or underspecified, can the model leverage its own parametric knowledge to supplement missing conditions during DAG construction and clarification?

---

> ### Author Response · Authors · 2025-11-23
>
> We sincerely appreciate your comments, suggestions, and every effort spent on reviewing our work. Here we attempt to address all your remaining concerns. In the following, we quote your comments and then give our detailed response point-by-point.
>
> > **W1: Baseline fairness**
>
> Clarify-DPO **does** have access to the retrieved documents. In our experimental setting, for ConditionalQA, our Synthetic dataset and ShARC, each query is paired with an annotated document. Each document is a policy description covering multiple sections such as *overview*, *eligibility*, and so on, with only some parts being directly relevant to the query. Since our study aims to examine a model's ability to **resolve ambiguity in the underspecified query once the relevant context has already been retrieved**, we provide each method with the annotated document corresponding to that query as its input context. This design ensures that all approaches, including Clarify-DPO, receive exactly the same contextual evidence, thereby maintaining fairness in comparison.
>
> > **W2: Training data parity**
>
> Regarding training-data parity, we ensure consistent and comparable training setups across methods. Our evaluation covers three benchmarks: **Synthetic**, **ConditionalQA**, and **ShARC**. Among all baselines, only Clarify-DPO and our method GPS require training. For our Synthetic dataset, we construct **Syn-Train / Syn-Test** using a 8:2 split. For ConditionalQA and ShARC, we evaluate on their respective public test sets (**ConditionalQA-Test** and **ShARC-Test**).
>
> For evaluation on **Syn-Test** and **ConditionalQA-Test**, both Clarify-DPO and GPS are trained **on Syn-Train**, ensuring identical training sources. For evaluation on **ShARC-Test**, Clarify-DPO is trained on the ShARC training split **ShARC-Train**, whereas GPS is not trained on ShARC and is evaluated directly using the checkpoint trained on **Syn-Train**. Performance highlights the **domain generalization capabilities** of our method GPS.
>
> > **W3: Efficiency evidence gap**
>
> We emphasize that our work focuses on **success-conditioned efficiency**, as the desired model behavior is to **clarify accurately before optimizing efficiency**.
>
> Under this perspective, **MCT alone is insufficient**, since it does not distinguish between successful and failed clarifications. Inspired by prior evaluation practice [1], we introduce **Weighted Clarification Turns (WCT)** to more appropriately capture efficiency under correctness:
>
> $WCT = p_{success} \times MCT_{success} + p_{failed} \times T_{\max}$
>
> where failed queries receive the maximum allowed clarification turns ( $T_{\max}=10$ in our experiment). Lower WCT indicates greater efficiency while preserving correctness. For completeness, we additionally evaluate **adaptive Clarify-DPO** and **adaptive BED-LLM**, which must autonomously decide whether clarification is needed. The results are shown below:
>
> **Synthetic (Qwen2.5-7B-Instruct)**
>
> | Method                   | SR ↑ | WCT ↓ | F1 ↑ |
> |--------------------------|------|--------|-------|
> | Base Method              | 21.2 | 7.88   | 0.0   |
> | ProCoT                   | 42.5 | 6.07   | 50.9  |
> | UoT                      | 32.8 | 7.05   | 89.2  |
> | BED-LLM                  | 40.9 | 6.41   | 100.0 |
> | **Adaptive BED-LLM**     | 34.6 | 6.89   | 95.2  |
> | Clarify-DPO              | 59.2 | 4.67   | 100.0 |
> | **Adaptive Clarify-DPO** | 32.6 | 7.04   | 96.2  |
> | **GPS (ours)**           | 60.2 | **4.59** | 96.4 |
>
> ---
>
> **ConditionalQA (Qwen2.5-7B-Instruct)**
>
> | Method                   | SR ↑ | WCT ↓ | F1 ↑ |
> |--------------------------|------|--------|-------|
> | Base Method              | 70.3 | 2.98   | 0.0   |
> | ProCoT                   | 71.6 | 2.95   | 10.4  |
> | UoT                      | 60.3 | 4.25   | 28.2  |
> | BED-LLM                  | 52.8 | 5.26   | 37.6  |
> | **Adaptive BED-LLM**     | 50.2 | 5.58   | 28.6  |
> | Clarify-DPO              | 72.0 | 3.52   | 37.6  |
> | **Adaptive Clarify-DPO** | 69.9 | 3.02   | 0.0   |
> | **GPS (ours)**           | 73.4 | **2.91** | 36.7 |
>
> **GPS achieves the best WCT on both backbones**, providing clear evidence of superior success-conditioned efficiency. The complete experimental results are provided in Table 1 of Section 5 in our updated manuscript.
>
> [1] Yokoyama N, Ha S, Batra D. Success Weighted by Completion Time: A Dynamics-Aware Evaluation Criteria for Embodied Navigation[C]. IROS, 2021.

---

> ### Author Response · Authors · 2025-11-23
>
> > **W4 & Q1: Domain generalization to domains where rules are fuzzier**
>
> To address the concern about generalization beyond rule-heavy policy datasets, we additionally evaluate GPS on **Abg-CoQA**[2] using LLaMA3-8B-Instruct as backbone. Abg-CoQA is an open-domain conversational QA dataset explicitly designed for **ambiguity detection and clarification in natural, fuzzy contexts**. Abg-CoQA covers ambiguity types such as coreference, event references, temporal uncertainty, and answer-type ambiguity, making it a suitable benchmark for evaluating transfer to **less structured, non-rule-based settings**. The results are shown below:
>
> | Method               | SR ↑ | WCT ↓| F1 ↑ |
> |----------------------|------|------|-------|
> | Base Method          |   46.9  | 5.31 |   0.0   |
> | ProCoT               | 43.0 | 5.78 | 17.6  |
> | Adaptive Clarify-DPO | 55.4 | 4.46 | 0.0   |
> | GPS (ours)           | **57.1** | **4.45** | **53.7**  |
>
> The results show that GPS demonstrates clear improvements over baseline methods across all metrics, indicating that **its structural reasoning ability transfers effectively to fuzzy, conversational environments**, even when conditional relationships are not explicitly rule-governed.
>
> [2] Guo M, Zhang M, Reddy S, et al. Abg-CoQA: Clarifying Ambiguity in Conversational Question Answering[C]. AKBC, 2021.

---

> > ### Author Response · Authors · 2025-11-27
> >
> > Dear Reviewer,
> >
> > I hope this message finds you well. We extend our gratitude once more for your valuable and insightful comments!
> >
> > We have provided careful and detailed responses to all your questions. It would be greatly appreciated if you could kindly let us know whether we have answered all your questions. Please also kindly let us know if you have any further questions, and we would like to try our best to resolve them before the deadline.
> >
> > Best regards, Authors of the paper 22850

---

> > > ### Comment · Reviewer_yNPb · 2025-11-27
> > >
> > > Thank you for your detailed response. All my concerns have been addressed. I will keep the positive score.

---

> > > > ### Author Response · Authors · 2025-11-27
> > > >
> > > > Dear Reviewer yNPb,
> > > >
> > > > Thank you for your time, effort, and positive assessment. Your guidance and insights have been invaluable in refining our research. We remain committed to addressing any remaining concerns and ensuring a better quality of our work.
> > > >
> > > > Once again, we are sincerely grateful for your kind guidance.
> > > >
> > > > Best regards, All authors of submission 22850

---

### Official Review · Reviewer_Zgt8 · 2025-11-01

**Soundness:** 2
**Presentation:** 1
**Contribution:** 3
**Rating:** 4
**Confidence:** 4

**Summary:**

The paper presents GPS, a two-stage framework for improving proactive clarification in retrieval-augmented generation (RAG) systems when user queries are ambiguous or underspecified. The method explicitly models conditional reasoning dependencies using a Directed Acyclic Graph (DAG) that captures logical structures across retrieved documents. The first stage constructs the DAG with theoretical guarantees of logical completeness, while the second stage performs dynamic traversal to prune inconsistent reasoning paths based on user feedback. To mitigate data scarcity, the authors propose a conditional path-guided data synthesis strategy and optimize DAG extraction using clarification-oriented reinforcement learning with hybrid rewards balancing accuracy and efficiency. Experimental results are conducted on Synthetic, ConditionalQA, and ShARC benchmarks.

**Strengths:**

+ GPS introduces a novel approach by integrating graph-based reasoning with proactive clarification in retrieval-augmented generation systems, which significantly advances the state of the art in handling underspecified queries.
+ The framework is supported by theoretical foundations, including formal guarantees of logical completeness for the constructed DAG, ensuring systematic and reliable reasoning.
+ Empirical results demonstrate that GPS outperforms baseline methods on Synthetic and conditional QA dataset on Success rate, and achieve comparable results on OOD dataset ShARC, showing generalization capabilities.

**Weaknesses:**

- The performance across the three benchmarks does not show clear and consistent superiority on all evaluation metrics, suggesting room for further improvement in robustness and overall gain.
- The methodological exposition could be clearer. the paper would benefit from more detailed algorithmic descriptions and richer examples to illustrate how conditional relationships are captured and resolved.
- The examples provided involve relatively simple condition–conclusion links; it remains unclear whether the DAG-based reasoning can handle more complex logical hierarchies, such as when a conclusion becomes a condition in a nested structure.
- Interestingly, on in-domain data, GPS exhibits only marginal gains and performs worse than Clarify-DPO in clarifier prediction accuracy. This suggests that the reasoner may struggle to interpret graph-based representations effectively when determining whether a query requires clarification. It would be valuable to explore enhancing the reasoner’s capability through joint training on synthesized DAG–QA pairs, allowing it to better align graph structures with clarification decisions.

**Questions:**

Since the improvement in Success Rate is only marginal, it would strengthen the paper to include qualitative comparison examples that clearly demonstrate the advantages of GPS in constructing higher-quality and more logically complete DAGs compared to existing methods. Such examples are essential to highlight why graph construction is both necessary and beneficial, beyond quantitative gains—showing how the proposed approach leads to more accurate condition–conclusion reasoning, better clarification paths, and enhanced interpretability relative to baselines.

---

> ### Author Response · Authors · 2025-11-23
>
> We sincerely appreciate your comments, suggestions, and every effort spent on reviewing our work. Here we attempt to address all your remaining concerns. In the following, we quote your comments and then give our detailed response point-by-point.
>
> > **W1: Performance across datasets is not uniformly superior on all metrics**
>
> We suspect the main concern lies in the fact that GPS does not consistently outperform all baselines on MCT and F1, despite consistent gains in SR. We argue that this inconsistency stems from **two fundamental issues in the original evaluation protocol**.
>
> **(1) Limitation of MCT: It does not measure the efficiency we truely care about.**
>
> MCT measures the average number of clarification turns but **does not reflect success-conditioned efficiency**, i.e., the desired behavior where a model should *first* clarify accurately and *then* minimize turns. Models that quickly reach incorrect conclusions can obtain low MCT despite poor reasoning quality, making the metric misaligned with proactive clarification objectives.
>
> Inspired by prior work[1], we introduce **Weighted Clarification Turns (WCT)**, which evaluates efficiency success-conditioned efficiency by penalizing failed clarifications with the maximum allowed clarification turns:
>
> $WCT = p_{success} \times MCT_{success} + p_{failed} \times T_{\max}$
>
> where failed queries receive the maximum allowed clarification turns ($T_{\max}=10$ in our experiment). Lower WCT indicates greater efficiency while preserving correctness.
>
> **(2) Limitation of F1: Evaluation setup affects F1 comparability.**
>
> For BED-LLM[2] and Clarify-DPO[3], we follow the original papers, where the model **always performs one clarification turn before answering**. In contrast, GPS and the other baselines must **decide whether clarification is needed first** and generate the clarification question if necessary. This asymmetric setup means Clarify-DPO and BED-LLM treats **all queries as ambiguous**, causing their F1 to be dominated by dataset label distribution rather than genuine ambiguity-detection ability.
>
> To enable a fairer and more rigorous comparison, we therefore evaluate **adaptive variants** of BED-LLM and Clarify-DPO, where the model must explicitly predict whether clarification is needed. The experimental settings for these adaptive baselines are detailed in Section 5.1 of the updated manuscript. The results are shown below:
>
> **Synthetic (Qwen2.5-7B-Instruct)**
>
> | Method                   | SR ↑ | WCT ↓ | F1 ↑ |
> |--------------------------|------|--------|-------|
> | Base Method              | 21.2 | 7.88   | 0.0   |
> | ProCoT                   | 42.5 | 6.07   | 50.9  |
> | UoT                      | 32.8 | 7.05   | 89.2  |
> | Adaptive BED-LLM         | 34.6 | 6.89   | 95.2  |
> | Adaptive Clarify-DPO     | 32.6 | 7.04   | 96.2  |
> | GPS (ours)               | **60.2** | **4.59** | **96.4** |
>
>
> **ConditionalQA (Qwen2.5-7B-Instruct)**
>
> | Method                   | SR ↑ | WCT ↓ | F1 ↑ |
> |--------------------------|------|--------|-------|
> | Base Method              | 70.3 | 2.98   | 0.0   |
> | ProCoT                   | 71.6 | 2.95   | 10.4  |
> | UoT                      | 60.3 | 4.25   | 28.2  |
> | Adaptive BED-LLM     | 50.2 | 5.58   | 28.6  |
> | Adaptive Clarify-DPO | 69.9 | 3.02   | 0.0   |
> | GPS (ours)           | **73.4** | **2.91** | **36.7** |
>
> **Conclusion**
>
> With a more principled efficiency metric (WCT) and adaptive baseline comparisons, our results consistently confirm the superiority of GPS in effectiveness, efficiency, and clarification prediction. We also provide additional qualitative analysis in the Appendix 5.4 for a deeper comparison.
>
> [1] Yokoyama N, Ha S, Batra D. Success Weighted by Completion Time: A Dynamics-Aware Evaluation Criteria for Embodied Navigation[C]. IROS, 2021.
>
> [2] Kobalczyk K, Astorga N, Liu T, et al. Active Task Disambiguation with LLMs[C]. ICLR, 2025.
>
> [3] Zhang M J Q, Knox W B, Choi E. Modeling Future Conversation Turns to Teach LLMs to Ask Clarifying Questions[C]. ICLR, 2025.
>
> > **W2: Methodological exposition could be clearer**
>
> To facilitate a better understanding of our method, we outline the complete pipeline as follows.
>
> During the inference process, we prompt the Reasoner extracts a reasoning DAG from the user query and the associated document. Based on the extracted DAG, the Clarifier performs dynamic traversal to identify missing condition values and interacts with the user simulator to collect the necessary information. After the traversal completes, the Clarifier generates the final answer using the gathered information.
>
> In our updated manuscript, we provide pseudo-code for the GPS inference procedure in Appendix D, the RL training algorithm for the Reasoner in Appendix E, and we include an illustrative example of the overall GPS information-seeking workflow in Appendix I.

---

> ### Author Response · Authors · 2025-11-23
>
> > **W3: Capability in handling complex, nested logical structures**
>
> From a theoretical standpoint, our method is fully capable of modeling nested logical structures. As established in **Proposition 1**, any finite conditional rule set can be mapped to a Directed Acyclic Graph (DAG): conjunctions correspond to sequential dependencies, disjunctions correspond to multi-parent nodes, and intermediate conclusions can recursively function as conditions for deeper branches. To demonstrate this in practice, we **provide a detailed example that contains nested logical structures in the Appendix H.2**. The resulting DAG shows that GPS can represent the nested logical structures and reason over such logical hierarchies.
>
> > **W4: Comparison with Clarify-DPO on F1 performance**
>
> As noted in our response to W1, Clarify-DPO treats all queries as ambiguous, making F1 performance largely reflect dataset label distribution rather than genuine ambiguity detection. To verify this, we conduct experiments with an adaptive Clarify-DPO variant that must explicitly decide whether clarification is required. The results are shown below:
>
> **Synthetic (Qwen2.5-7B-Instruct)**
>
> | Method                   | SR ↑ | WCT ↓ | F1 ↑ |
> |--------------------------|------|--------|-------|
> | Base Method              | 21.2 | 7.88   | 0.0   |
> | **Clarify-DPO**              | **59.2** | 4.67   | **100.0** |
> | **Adaptive Clarify-DPO** | **32.6** | 7.04   | **96.2**  |
> | GPS (ours)           | 60.2 | 4.59 | 96.4 |
>
> **ConditionalQA (Qwen2.5-7B-Instruct)**
>
> | Method                   | SR ↑ | WCT ↓ | F1 ↑ |
> |--------------------------|------|--------|-------|
> | Base Method              | 70.3 | 2.98   | 0.0   |
> | **Clarify-DPO**              | **72.0** | 3.52   | **37.6**  |
> | **Adaptive Clarify-DPO** | **69.9** | 3.02   | **0.0**   |
> | GPS (ours)          | 73.4 | 2.91 | 36.7 |
>
> As shown by these results, the F1 of Adaptive Clarify-DPO drops on both datasets, especially on ConditionalQA. This confirms that ambiguity detection is inherently difficult and that Clarify-DPO's high F1 stems from its always-ask behavior rather than true reasoning capability.
>
> In our method GPS, ambiguity detection is performed during dynamic traversal by the Clarifier, rather than by the Reasoner. At each condition node, the Clarifier determines whether the condition value can be inferred from the user query; if not, the Clarifier deems clarification necessary and issues a question. Thus, ambiguity detection emerges from unresolved condition values encountered along the traversal path, rather than from any explicit graph-level classification.
>
> We appreciate your suggestion of enhancing the Reasoner through joint training on synthesized DAG–QA pairs. Although ambiguity detection in GPS are not from explicit graph-level classification, aligning DAG construction with ambiguity behavior is indeed an interesting direction. We will include this point as a promising avenue for future work.
>
> > **Q1: Qualitative analysis comparing GPS with baseline methods**
>
> In our updated manuscript, we provide qualitative analysis in **Section 5.4 and Appendix H.1**. The qualitative comparison examples show that our method GPS leads to more accurate condition–conclusion reasoning, better clarification paths, and enhanced interpretability relative to baselines.
>
> ---

---

> > ### Author Response · Authors · 2025-11-27
> >
> > Dear Reviewer,
> >
> > I hope this message finds you well. We extend our gratitude once more for your valuable and insightful comments!
> >
> > We have provided careful and detailed responses to all your questions. It would be greatly appreciated if you could kindly let us know whether we have answered all your questions. Please also kindly let us know if you have any further questions, and we would like to try our best to resolve them before the deadline.
> >
> > Best regards, Authors of the paper 22850

---

### Meta-Review · Area_Chair_ZV5z · 2026-01-08

**Summary:**

This paper addresses the issue of ambiguity in retrieval-augmented generation (RAG) by training language models (LLMs) to proactively ask more effective clarifying questions. It introduces GPS, a two-step methodology: initially, it represents the retrieved knowledge using a Directed Acyclic Graph (DAG), allowing the model to logically reason through conditional rules comprehensively. Next, it interactively traverses and prunes this graph based on user responses to maintain efficient clarification. To implement this approach in practice, the authors create training data that captures various conditional paths and further refine the model using a clarification-focused reinforcement learning objective that balances both effectiveness and efficiency.

**Reviewer Concerns:**

The performance is not consistent on all datasets.
The fairness of the baselines should be improved.
The methodological exposition could be clearer.

**Reviewer Scores:**

The scores of the reviewers are 4,6,6,2, and I believe the authors have addressed most of the reviewers' concerns.

---

### Decision · Program_Chairs · 2026-01-26

Accept (Poster)